# Live imaging of hair bundle polarity acquisition demonstrates a critical timeline for transcription factor Emx2

Yosuke Tona[†], Doris K Wu*

National Institute on Deafness and Other Communication Disorders, National Institutes of Health, Bethesda, United States

**Abstract** Directional sensitivity of hair cells (HCs) is conferred by the aymmetric apical hair bundle, comprised of a kinocilium and stereocilia staircase. The mother centriole (MC) forms the base of the kinocilium and the stereocilia develop adjacent to it. Previously, we showed that transcription factor Emx2 reverses hair bundle orientation and its expression in the mouse vestibular utricle is restricted, resulting in two regions of opposite bundle orientation (Jiang et al., 2017). Here, we investigated establishment of opposite bundle orientation in embryonic utricles by live-imaging GFP-labeled centrioles in HCs. The daughter centriole invariably migrated ahead of the MC from the center to their respective peripheral locations in HCs. Comparing HCs between utricular regions, centriole trajectories were similar but they migrated toward opposite directions, suggesting that Emx2 pre-patterned HCs prior to centriole migration. Ectopic *Emx2*, however, reversed centriole trajectory within hours during a critical time-window when centriole trajectory was responsive to Emx2.

*For correspondence:
wud@nidcd.nih.gov

Present address: [†]Department of Otolaryngology - Head and Neck Surgery, Graduate School of Medicine, Kyoto University, Sakyo-ku, Kyoto, Japan

## Introduction

The mammalian inner ear comprises six major sensory organs including the cochlea, two maculae and three cristae. The cochlea detects sound, whereas the maculae and cristae detect linear accelerations and angular velocity of head movements, respectively. Each sensory organ consists of sensory hair cells (HCs), which are the mechano-transducers of sound and head movements, and each HC is surrounded by supporting cells. Erected on the apical surface of HCs is the stereociliary bundle/hair bundle, which is comprised of a stereociliary staircase that is tethered to the tallest rod of the bundle, the kinocilium. When the hair bundle is deflected toward the kinocilium, the mechano-transducer channels on the tips of the stereocilia open, which allow entry of positive ions and activation of the HC (*Shotwell et al., 1981*). Thus, orientation of the hair bundle provides the directional sensitivity of its HC.

Each sensory organ of the inner ear exhibits a defined hair bundle orientation pattern among HCs. Unlike other sensory organs in which hair bundles are unidirectional, the macula of the utricle and saccule exhibit an opposite bundle orientation pattern. Each macula can be divided by a line of polarity reversal (LPR) into two regions, across which the hair bundles are arranged in opposite orientations (*Figure 1A*; *Flock, 1964*). Although proper alignment of the hair bundles in sensory organs requires the Wnt signaling pathway and the core planar cell polarity pathway (*Dabdoub et al., 2003*; *Goodrich and Strutt, 2011*; *Jones et al., 2014*; *Lee et al., 2012*; *Montcouquiol et al., 2003*; *Montcouquiol et al., 2006*), the LPR in the maculae is generated by the transcription factor Emx2. The restricted expression of Emx2 to one side of the LPR causes hair bundles within to reverse their orientation by 180˚ (*Figure 1A*, green color; *Jiang et al., 2017*; *Holley et al., 2010*). In *Emx2* knockout or gain-of-function utricles, the LPR is absent and all hair bundles are unidirectional (*Figure 1B*).

Largely based on scanning electron microscopy and immunostaining results, it is thought that the hair bundle is established by first docking of the mother centriole (MC) to the apical center of a nascent HC, where the MC forms the base of the kinocilium (*Lu and Sipe, 2016*). Then, the kinocilium is relocated from the apical center of the HC to the periphery (*Figure 1C*). After the kinocilium acquires its final position, the stereocilia staircase is gradually built next to the kinocilium (*Lu and Sipe, 2016*; *Tarchini et al., 2013*; *Cotanche and Corwin, 1991*; *Tilney et al., 1992*). The role of the

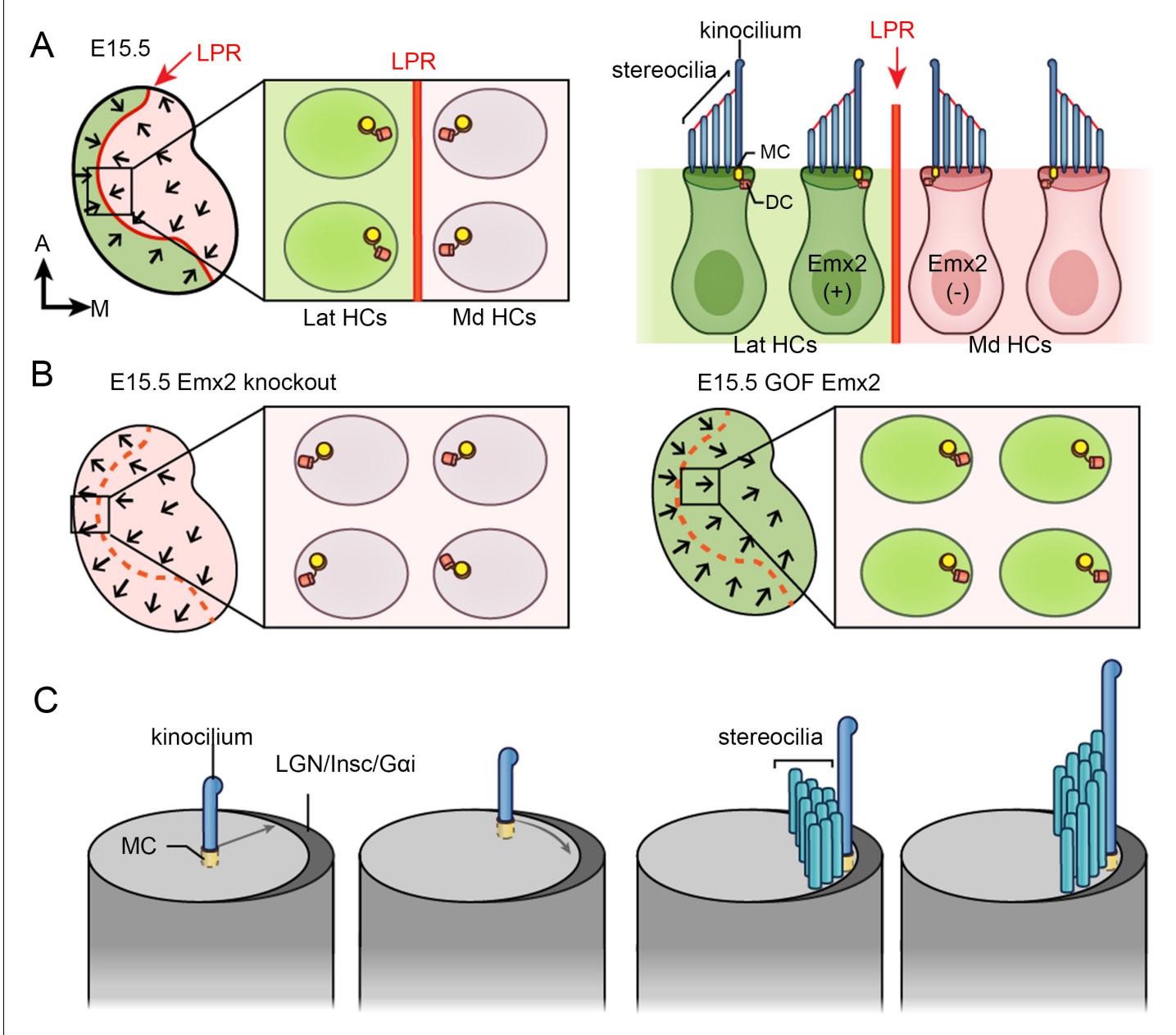

**Figure 1.** Hair bundle orientation establishment in the developing mouse utricle. (**A**) In E15.5 utricle, hair bundles are pointing toward each other (arrows) across the line of polarity reversal (LPR, red). The kinocilium is located asymmetrically at the lateral region of the apical hair cell (HC) surface. The mother centriole (MC) (yellow), which forms the basal body of the kinocilium, is located more centrally relatively to the daughter centriole (DC) (orange). Emx2-positive domain is in green. (**B**) Schematics showing hair bundle orientation in *Emx2* knockout and gain-of-Function utricles. (**C**) Model showing asymmetric hair bundle establishment requires the LGN/Inscuteable/Gαi complex. Orientations: A, anterior; M, medial.

daughter centriole (DC), the inherent partner of the MC, during hair bundle establishment is not known but it is located slightly more peripheral and basal to the MC in mature HCs (*Figure 1A*; *Sipe and Lu, 2011*).

Emx2 has a conserved role in reversing hair bundle orientation in HCs of mice and zebrafish (*Jiang et al., 2017*). However, the timing of Emx2 required to mediate hair bundle reversal is not clear. In other tissues such as the brain, olfactory epithelium and urogenital system, *Emx2* is suggested to function as a patterning gene since the lack of *Emx2* affects regional formation of these tissues (*Miyamoto et al., 1997*; *Pellegrini et al., 1996*). Thus, Emx2 could have a similar role in patterning the lateral utricle or specifying the fate of HCs, which indirectly leads to hair bundle reversal. However, Emx2 is known to require the LGN/Insc/Gαi complex in mediating hair bundle reversal (*Jiang et al., 2017*). The LGN/Insc/Gαi complex forms an asymmetrical crescent on the apical surface of cochlear HCs and this complex is important for guiding the kinocilium to its proper location for cochlear hair bundle establishment and for subsequent stereocilia staircase formation (*Figure 1C*; *Ezan et al., 2013*; *Tarchini et al., 2013*; *Tarchini et al., 2016*). Therefore, regardless of the mechanisms or timing, Emx2 executes hair bundle reversal by guiding centriole positioning.

In this study, we investigated the timing of Emx2 in hair bundle reversal. We first live-imaged GFP-labeled centrioles in nascent utricular HCs to track the process of hair bundle polarity acquisition. Then, we compared centriole migration trajectories between the Emx2-positive HCs in the lateral and Emx2-negative HCs in the medial utricle to determine whether there is a fundamental difference in their hair bundle polarity establishment (*Figure 2B*). We found that there were no obvious differences between medial and lateral HCs in the centriole migration pattern, other than their opposite direction of trajectory. These results indicate that Emx2 has pre-patterned the HCs prior to centriole migration. However, ectopic *Emx2* in naive medial utricular HCs demonstrated that Emx2 can alter predetermined centriole trajectories within 12 hours (hr) and there is a critical time window for centrioles to respond to Emx2. Furthermore, our live-imaging results showed dynamic relationships between the MC and DC, suggesting that the DC may have an active role in guiding the MC. Disruption of microtubule experiments indicate that both centrioles are actively being translocated to its peripheral location via the microtubule network, and ninein, a centrosomal protein, may anchor microtubules to facilitate centriole migration.

## Results

### Migration of DC precedes MC in medial utricular HCs during hair bundle establishment

To address how hair bundles are established across the LPR in the macular organs, we first live-imaged centriole movements as a proxy for hair bundle orientation establishment in medial utricular HCs (Md HCs) of *Atoh1^{Cre}*; *Rosa^{tdT/+}*; *CAG:GFP-Centrin2* mice at embryonic day (E) 13.5, in which all centrioles are GFP-positive and nascent HCs are tdTomato-positive (*Figures 2A,C–E* and *3A–C*). At this stage, most of the nascent HCs are located in the Emx2-negative, medial region of the utricle and few are in the Emx2-positive, lateral region (*Figure 2A*; *Jiang et al., 2017*; *Yang et al., 2017*). Some of the Md HCs are already polarized with the kinocilium asymmetrically located at the lateral periphery of the apical surface, whereas other immature HCs show the kinocilium at the center of the apical surface (*Figure 2A*, *Figure 2—figure supplement 1*). In the course of our experiments, we tracked centriole movements in a total of 32 nascent Md HCs for 40.8 hr. *Figures 2* and *3* illustrate the locations of two of these HCs (*Figure 2C*, Md HC1, *Figure 3A*, Md HC2) and their centriole movements (*Figures 2D–E* and *3B–C*). The trajectories of centriole movements (*Figure 2D–D′* and *Figure 3B–B′*) and selected frames of the time-lapse recordings (*Figures 2E* and *3C*) are shown. The identity of the MC (*Figures 2D–E* and *3B–C*, *Figure 2—figure supplement 1*, yellow color) was determined based on its more apical location within the HC than the DC (*Figures 2D–E* and *3B–C*, *Figure 2—figure supplement 1*, orange color; *Sipe and Lu, 2011*). The identity of the MC was further validated by its association with the kinocilium marker, Arl13b (*Figure 2—figure supplement 1*).

Based on tracking and analyses of trajectories and relationships between the MC and DC in Md HCs, two phases of centriole movements emerged (*Figure 2C–E*, *Figure 2—figure supplement 2*, *Figure 2—video 1*). In Phase I, the MC (yellow arrowhead) was positioned near the center of HC's

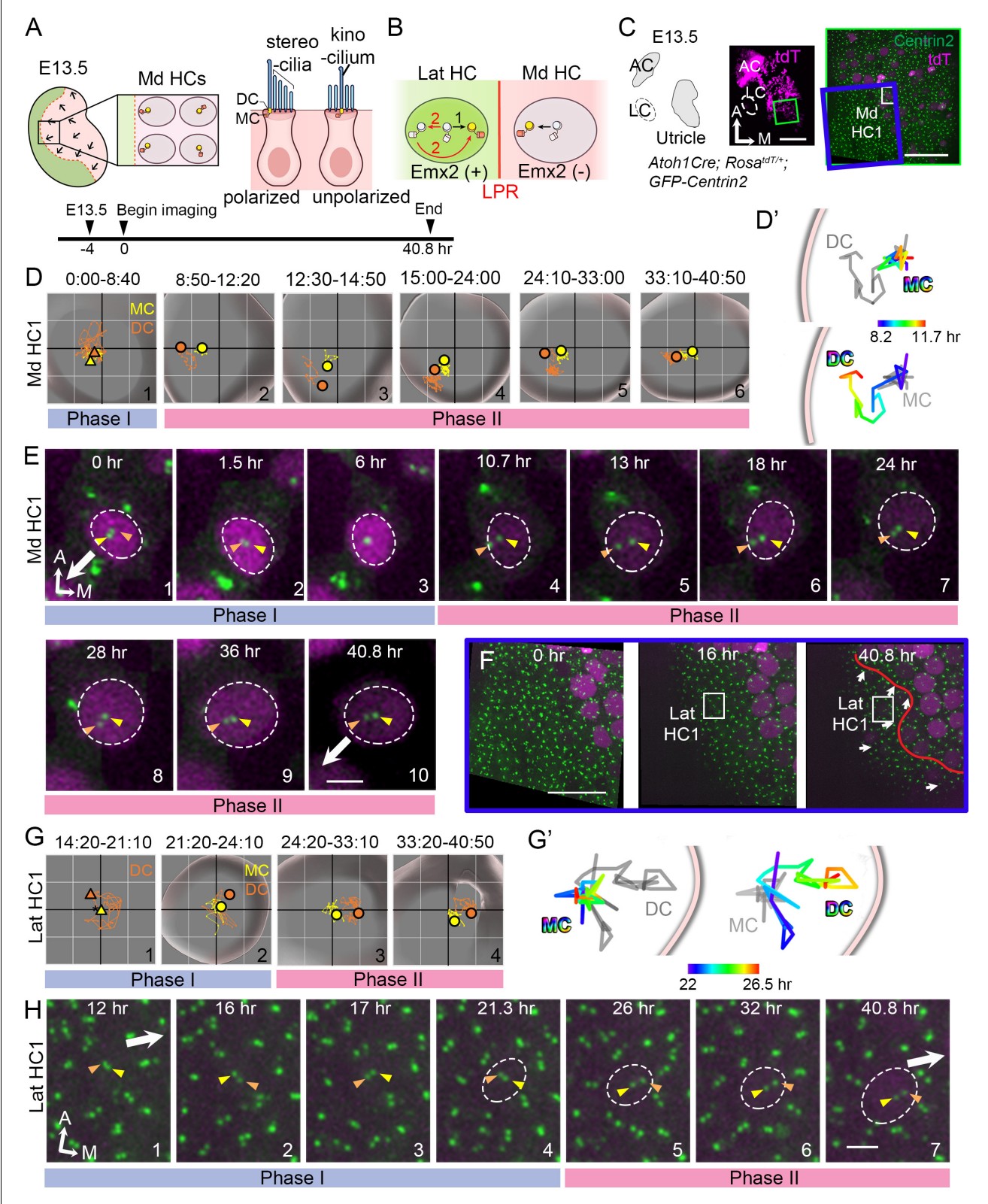

**Figure 2.** Live imaging of hair bundle establishment in medial and lateral hair cells (HCs) based on centriole movements. (**A**) Schematic of E13.5 utricle. The line of polarity reversal (LPR) is not apparent (dotted line) since HCs are mostly absent in the Emx2-positive lateral region (green) at this age. In the medial utricle, while some HCs (Md HCs) are polarized showing the kinocilium located asymmetrically at the lateral region, others are immature and unpolarized with the kinocilium located at the center. (**B**) The mother centriole (MC)/kinocilium in a lateral HC (Lat HC) could migrate directly towards

*Figure 2 continued on next page*

*Figure 2 continued*

the medial side (black arrow, #1) or it could first migrate toward the lateral side before reversing its direction to the medial side (red arrows, #2). (**C**) Schematic drawing and images of an *Atoh1^Cre^; Rosa^tdT/+^; CAG:GFP-Centrin2* utricular explant at E13.5 showing the location of Md HC1 (small white rectangle). (**D–E**) Centriole trajectories (**D, D'**) and selected time frames of apical views (**E**) of MC (yellow) and daughter centriole (DC) (orange) in Md HC1 from time-lapse recording (***Figure 2—source data 1***). (**D**) Yellow and orange triangles represent the beginning positions of respective MC and DC in each time period, and the circled dots represent the final positions in each time period. The yellow and orange lines represent trajectories of the respective MC and DC and are plotted relative to the center of the HC, which is represented by the centroid of the graph. Each small grid is 1.25 × 1.25 μm. The apical surface of the HC is shown in light grey and the white rim marks the width of the HC body (data extracted using Imaris software). All subsequent live-imaging graphs are organized in a similar manner. Initially, the DC is moving vigorously around the MC (Phase I), then the DC starts to move towards the periphery, which is followed by the MC (Phase II). (**D'**) Selected temporal trajectories (color-coded) of centrioles in Md HC1 from the end of Phase I to the beginning of Phase II (8:20-11:40 hr), showing the initial movements of DC toward the periphery ahead of the MC. The pink line indicates the edge of the apical HC surface. (**E**) In Phase I (blue bar), the basal body/MC (yellow arrowhead) is located at the center of the HC, whereas the DC (orange arrowhead) moves around the MC. In Phase II (pink bar), the DC starts to migrate toward the lateral periphery of the HC (#4-#5). This migration is followed by the MC (#6-#7). Then, both centrioles move towards the center as a pair (#8-#10). The white arrow represents the direction where the hair bundles should be pointing in this region of the utricle. (**F–H**) Lat HC1. (**F**) Time frames of the blue rectangle area in (**C**) at 0 hr, 16 hr and 40.8 hr of the recording showing the gradual appearance of tdTomato signals in the lateral utricle and the location of the Lat HC1. Arrows represent hair bundle orientation. (**G–H**) Total (**G**) and selected temporal (**G'**) trajectories as well as selected time frames of the MC and DC (**H**) in the Lat HC1 (***Figure 2—source data 1***). TdTomato signal in Lat HC1 was not detectable until 21 hr into the recording (F, H#1–3), which made it difficult to identify the center of the HC. Therefore, the position of the MC (yellow triangle) was used as a proxy for the center of the HC (asterisk) for #1 in (**G**) until the center of HC can be determined in #2–4 in (**G**). Lat HC1 shows similar centriole movements as Md HC1 with DC precedes MC to the periphery (**G'**). AC, anterior crista; LC, lateral crista; Md HC, medial utricular HC; Lat HC, lateral utricular HC. Scale bars: 100 μm (low magnification) and 30 μm (high magnification) in (**C**), 30 μm in (**F**), 3 μm in (**E**) and (**H**).

The online version of this article includes the following video, source data, and figure supplement(s) for figure 2:

**Source data 1.** Coordinates of centriole positions relative to the center of the HC or the MC in Md HC1 and Lat HC1.

**Figure supplement 1.** Identification of mother and daughter centrioles in polarized and unpolarized medial utricular hair cells (Md HCs).

**Figure supplement 2.** Traveled distance and speed between the mother centriole (MC) and daughter centriole (DC) in medial utricular hair cells (Md HCs).

**Figure supplement 2—source data 1.** The x-y moving speeds of centrioles in Md HC1 and Md HC2 during centriole migration.

**Figure supplement 2—source data 2.** The x-y distance between centrioles in Md HC1 and Md HC2 during centriole migration.

**Figure supplement 2—source data 3.** The z-distance between centrioles in Md HC1 and Md HC2 during centriole migration.

**Figure supplement 3.** Live imaging of hair bundle establishment in lateral utricular hair cells (Lat HCs) based on centriole movements.

**Figure supplement 3—source data 1.** Coordinates of centriole positions relative to the center of the hair cells (HC) or the mother centriole (MC) in Lat HC2.

**Figure supplement 3—source data 2.** The x-y distance between centrioles in Lat HC1 and Lat HC2.

**Figure supplement 3—source data 3.** The z distance between centrioles during migration in Lat HC1 and Lat HC2.

**Figure supplement 3—source data 4.** The x-y moving speeds of centrioles during migration in Lat HC1 and Lat HC2.

**Figure supplement 4.** Absence of polarizing centriole trajectories in medial and lateral supporting cells.

**Figure supplement 4—source data 1.** Coordinates of daughter centriole (DC) positions relative to the mother centriole (MC) in Md and Lat supporting cells.

**Figure supplement 4—source data 2.** Relative positions of the daughter centriole (DC) compared to the mother centriole (MC) in Md and Lat supporting cells at different time points.

**Figure supplement 5.** Expression of *Emx2* in the developing utricle.

**Figure 2—video 1.** Md HC1 and Md HC2.
https://elifesciences.org/articles/59282#fig2video1

**Figure 2—video 2.** Lat HC1 and Lat HC2.
https://elifesciences.org/articles/59282#fig2video2

apical surface, whereas the DC (orange arrowhead) moved rapidly and sporadically around the MC (***Figure 2D***#1, 2E#1–3, 3B#1, 3C#1–3). The x-y and z distances between the two centrioles were variable during Phase I (***Figure 2—figure supplement 2, A,C***), in which the two centrioles could be far apart (***Figure 3C***#1) or aligned along the apical-basal axis (***Figure 2E***#3, 3C#3). The speed of the DC was also faster, up to 1.2 μm per a 10-min time frame (orange), whereas the speed of the MC (yellow) was more stable, not exceeding 0.5 μm per time frame after correcting for HC drift (***Figure 2—figure supplement 2B***, Md HC1: p=2.0×10$^{-8}$, Md HC2: p=0.015, ***Figure 2—figure supplement 2—source data 1***). By contrast, Phase II was characterized by the DC showing directional

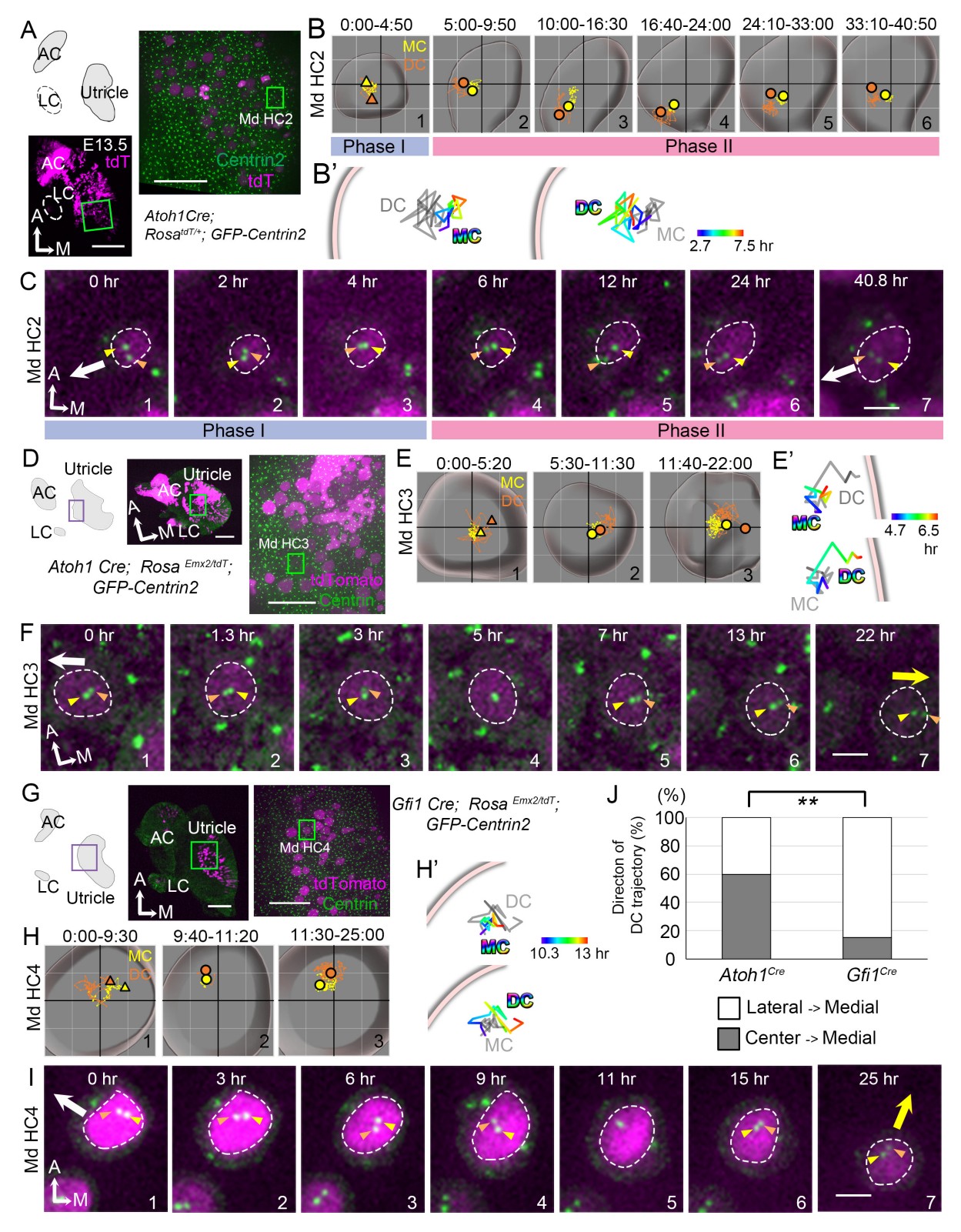

**Figure 3.** Trajectories of centriole movements in *Emx2* gain-of-function Md HCs. (A–C) Schematic drawing and images of Md HC2 in *Atoh1^Cre^; Rosa^tdT/+^; CAG:GFP-Centrin2* control utricle at E13.5 (A). (B–C) Total (B) and temporal (B') trajectories as well as selected frames (C) from a recording of mother centriole (MC) (yellow) and daughter centriole (DC) (orange) in Md HC2 (*Figure 3—source data 1*). Trajectory is similar to Md HC1 in *Figure 2*. Briefly, the DC is moving vigorously around the MC (Phase I), then the DC starts to move toward the periphery, where hair bundles should be pointing

*Figure 3 continued on next page*

Figure 3 continued

in this region of the utricle (white arrow in C). This trajectory is followed by the MC (Phase II). (D–F) Schematic drawing and low- and high-magnification images of Md HC3 in *Atoh1^{Cre}; Rosa^{Emx2/tdT}; CAG:GFP-Centrin2* utricle at E13.5 (D). Total (E) and selected temporal (E') trajectories as well as selected apical views (F) of MC (yellow) and DC (orange) in Md HC3 over-expressing *Emx2* (*Figure 3—source data 1*). (F) The DC (orange arrowhead) moves around the MC (yellow arrowhead) in the center of the apical HC surface (#1-4). Then, the DC starts to move toward the medial side (yellow arrow), which is followed by the MC (#5-7). (G–I) Schematic drawing, low- and high-magnification images of Md HC4 in *Gfi1^{Cre}; Rosa^{Emx2/tdT}; CAG:GFP-Centrin2* utricular explant (G). Total (H) and selected temporal (H') trajectories as well as selected apical views (I) of the MC and DC in Md HC4 over-expressing *Emx2* (*Figure 3—source data 1*). (H) Between 0:00 and 9:30 hr (#1), the centrioles are migrating toward the lateral side of the HC with the DC more lateral than the MC. In #2 (9:40-11:20 hr), the DC starts to change its position to the medial side of the MC, which becomes more apparent in #3 (11:30-25:00 hr). (H') Temporal trajectories of centrioles in Md HC4 from 10:20 to 13:00 hr showing the DC in the periphery moving medial to the MC toward the center of the HC. (I) In panels 1–4, the DC (orange arrowhead) is heading toward the lateral side of the utricle (white arrow), then it changes course and moves to the medial side of MC (panels 6–7, yellow arrow). (J) Percentages of DC with two different trajectories in HCs ectopically expressing *Emx2* using either *Atoh1^{cre}* or *Gfi1^{cre}*. Total number of HCs analyzed: *Atoh1^{cre}*, n = 15; *Gfi1^{cre}*, n = 39 (*Figure 3—source data 2*). Scale bars: 100 μm (low mag) and 30 μm (high mag) in (A), (D) and (G), 3 μm in (C), (F) and (I). **p<0.01.

The online version of this article includes the following video, source data, and figure supplement(s) for figure 3:

**Source data 1.** Coordinates of centriole positions relative to the center of the cell in Md HC2, Md HC3, and Md HC4.

**Source data 2.** Quantification of daughter centriole trajectories in *Atoh1^{Cre}; Rosa^{Emx2/tdT}; CAG:GFP-Centrin2* and *Gfi1^{Cre}; Rosa^{Emx2/tdT}; CAG:GFP-Centrin2* utricles.

**Figure supplement 1.** Gain-of-function of *Emx2* reverses hair bundle orientation in Md HCs of *Atoh1^{Cre}; Rosa^{Emx2/tdT}; CAG:GFP-Centrin2* utricles.

**Figure supplement 2.** Centriole trajectories in *Emx2* gain-of-function medial hair cells (Md HCs) using *Gfi1^{Cre}*.

**Figure supplement 2—source data 1.** Coordinates of centriole positions relative to the center of the HC in Md HC5, Md HC6, and Md HC7.

**Figure 3—video 1.** Md HC3 and Md HC4.

https://elifesciences.org/articles/59282#fig3video1

movements toward the lateral utricle (*Figure 2D*#2, 2D', 2E#4–5, 3B#2, 3B', 3C#4) where the kinocilium of Md HCs will subsequently reside in the lateral periphery (white arrow). Then, the MC migrated towards the direction of the DC (*Figure 2D*#3–4, 2E#6–7, 3B#3–4, 3C#5–6). The DC continued to move faster than the MC in Phase II (*Figure 2—figure supplement 2B*, Md HC1: p=1.8×10^{-6}, Md HC2: p=4.7×10^{-6}, *Figure 2—figure supplement 2—source data 1*) and the average x-y distance between the two centrioles in Phase II was significantly larger than Phase I but the z-distance was smaller in Phase II than Phase I (*Figure 2—figure supplement 2A,C*, Md HC1: x-y distance, p=1.9×10^{-10}, *Figure 2—figure supplement 2—source data 2*, z-distance, p=7.0×10^{-6}, Md HC2: x-y distance, p=2.0×10^{-5}, z-distance, p=1.0×10^{-5}, *Figure 2—figure supplement 2—source data 3*). Between 24 to 40.8 hr of recording, the two centrioles were observed to relocate again towards the center of HC as a pair and maintaining the relative positions between each other (*Figure 2D*#5–6, 2E#7–10, 3B#5-#6, 3C#6–7). This movement resembled what was previously described for the kinocilium being shifted centrally due to formation of the bare zone comprised of the LGN/Insc/Gαi complex (*Tarchini et al., 2013*).

## Lateral HCs show trajectories of MC and DC similar to medial HCs

Hair bundles in the lateral utricle are in opposite orientation from the default hair bundles in the medial utricle (*Figure 1A*). We investigated whether centrioles in the Emx2-positive lateral HCs (Lat HCs) migrate directly to their destined position in the medial periphery or they first migrate to the lateral position before relocating to the medial destined position (*Figure 2B*). Since most of the HCs in the lateral utricle initiate terminal mitosis at E14.5 or later (*Jiang et al., 2017*), live-imaging of E13.5 utricular explants was analyzed for 41 hr (*Figure 2F–H*, *Figure 2—figure supplement 3, A–E*). At the beginning of recordings, tdTomato-positive HCs were only found in the medial utricle. Sixteen hr into imaging, several HCs emerged in the lateral region of the utricle and their centrioles were positioned in the medial periphery of the apical surface of the HCs by the end of the recording (*Figure 2F*, Lat HC1, *Figure 2—figure supplement 3A*, Lat HC2). Based on the positions of HCs at the end of the recordings, we re-traced and analyzed the centriole trajectories of 9 Lat HCs. Trajectories of centrioles in Lat HCs can also be grouped into two phases similar to Md HCs. In Phase I, the DC moved sporadically around the MC before or soon after the tdTomato signal was evident (*Figure 2G*#1–2, 2H#1–4, *Figure 2—figure supplement 3, B*#1–2, C#1–4, *Figure 2—video 2*). Then, in Phase II, the DC started to migrate towards the medial side, which was followed by the

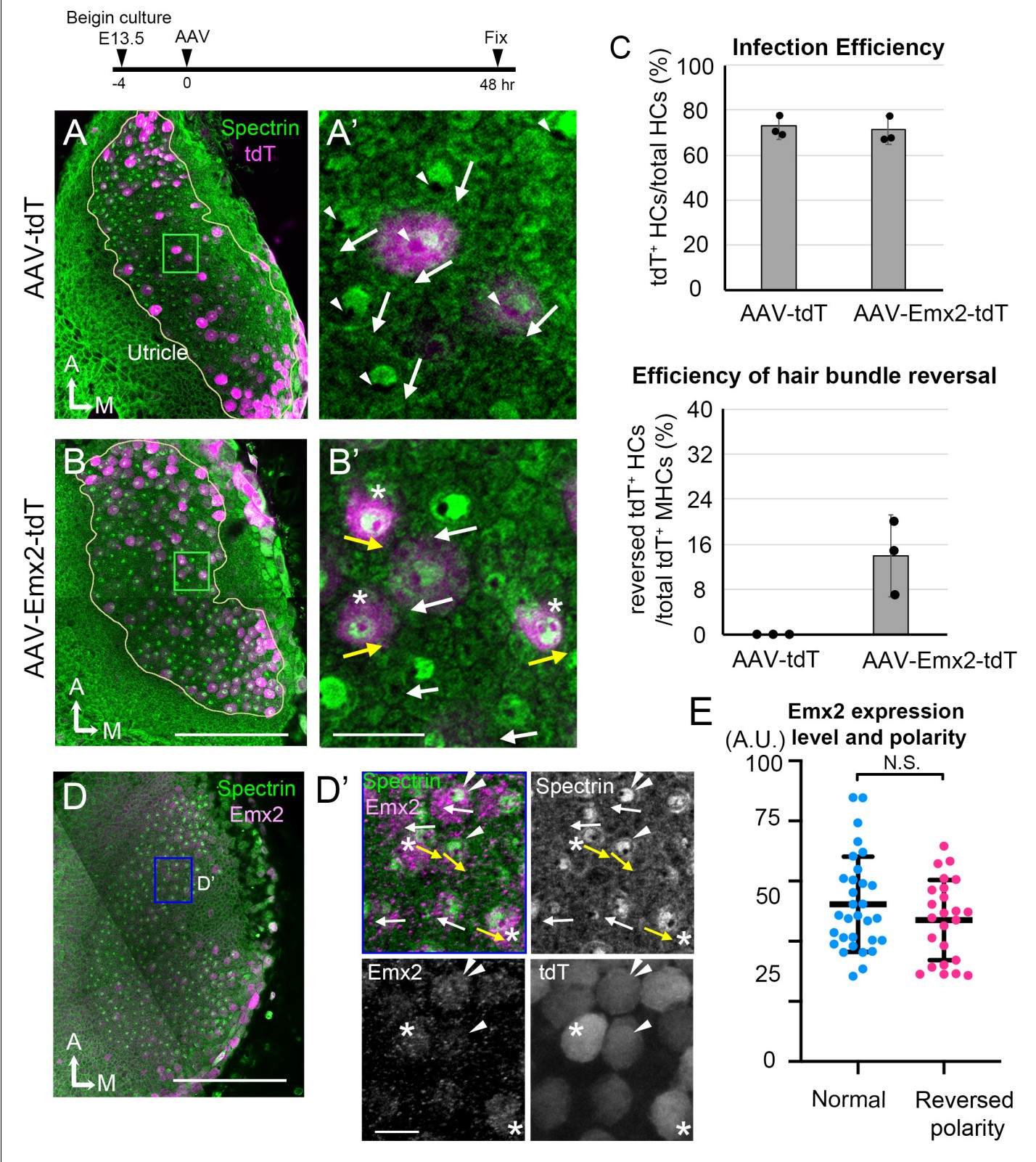

**Figure 4.** AAV-Emx2 infection alters hair bundle orientation in medial hair cells (Md HCs). (A, A') Low- (A) and high-magnification (A') images of the rectangular area of a control explant infected with AAV-tdT (tdT: magenta) and stained with anti-β2-spectrin antibodies (green), in which the absence of staining indicates the kinocilium location (arrowhead) and HC orientation (arrow). The two AAV-tdT infected HCs (magenta) show hair bundle

*Figure 4 continued on next page*

Figure 4 continued

orientation similar to non-infected HCs. (B,B') Low- (B) and high-magnification (B') images of an utricular explant infected with AAV-Emx2-tdT (tdT: magenta), stained with anti-β2-spectrin antibodies (green). (B') Some of the AAV-Emx2-tdT infected Md HCs show opposite hair bundle orientation (yellow arrow and asterisk) from the rest of the non-infected HCs (white arrows). (C) Efficiency of viral infections and efficiency of hair bundle reversal among infected HCs (n = 3 experiments for each condition). (D–D') Low (D) and high magnification (D') images of an utricular explant infected with AAV-Emx2-tdT and stained with anti-Emx2 (magenta) and β2-spectrin antibodies (green). (D') Four HCs with normal bundle orientation (white arrows) showing variable Emx2 staining including, one with strong Emx2 expression (white arrow with double arrowheads). The three HCs with abnormal bundle orientation (yellow arrows pointing towards the medial) show one with weak (arrowhead) and two with strong (asterisks) Emx2 expression. (E) Quantification of Emx2 immunoreactivity among infected HCs with normal (n = 30) or reversed bundle orientation (n = 23). There is no significant difference in the levels of Emx2 expression between the two groups of HCs. Error bars represent SD. Scale bars: 100 μm in (B) and applies to (A) and 10 μm in (B') and applies to (A'). 100 μm in (D) and 10 μm in (D').

The online version of this article includes the following source data for figure 4:

**Source data 1.** Efficiency of infections and hair bundle reversal phenotype mediated by control (AAV-tdT) and AAV-Emx2-tdT viruses.
**Source data 2.** Correlation of Emx2 expression levels with hair bundle orientation.

MC (*Figure 2G*#3–4, 2G', 2H#5–7, *Figure 2—figure supplement 3B*#3–4, B', C#5–7). Similar to Md HCs, the average x-y and z distances between the two centrioles were variable but the x-y distance was closer in Phase I than Phase II, but the z-distance was further apart in Phase I than Phase II (*Figure 2—figure supplement 3, D,F*, Lat HC1: x-y distance, p=2.7×10$^{-11}$, *Figure 2—figure supplement 3—source data 2*, z–distance, p=3.0×10$^{-4}$, *Figure 2—figure supplement 3—source data 3*, Lat HC2: x-y distance, p=1.7×10$^{-17}$, z-distance, p=3.8×10$^{-3}$). Average moving speed of the DC was faster than that of the MC in each phase, although three out of nine Lat HCs failed to show a significant difference in speed between the two centrioles in Phase II (*Figure 2—figure supplement 3E*, Lat HC1: Phase I p=0.0078, Phase II p=0.028, Lat HC2: Phase I p=0.048, Phase II p=0.17, *Figure 2—figure supplement 3—source data 4*). These centriole trajectories in HCs were not observed in supporting cells in either the medial or lateral utricle (*Figure 2—figure supplement 4*, *Figure 2—figure supplement 4—source data 1*, *Figure 2—figure supplement 4—source data 2*). Centrioles in supporting cells behaved in a similar manner as centrioles in Phase I HCs, with the DC showing an inconsistent position in relationship to the MC.

Thus far, our analyses revealed that Md HCs and Lat HCs show similar centriole trajectories in reaching their destinations in opposite sides of the HC. The two centrioles in the Lat HC moved directly toward the medial periphery (*Figure 2B*#1) without first reaching the lateral periphery like in the Md HC and then relocated to the medial periphery (#2). These results indicate that Emx2 has already exerted its effects on the Lat HCs prior to hair bundle establishment. Notably, *Emx2* transcripts are detected in the lateral region at E11.5, three days ahead of HC formation that begins at E14.5 (*Figure 2—figure supplement 5*; *Jiang et al., 2017*). Taken together, these results suggest that the mechanism of Emx2 in altering hair bundle orientation in the lateral utricle could be indirect via possible regional patterning and/or HC fate determination.

## Reversing hair bundle orientation by ectopic *Emx2*

To gain further insight into the mechanism of Emx2 in altering hair bundle orientation, we investigated the time it takes for ectopic Emx2 to reverse hair bundle orientation in Md HCs (*Jiang et al., 2017*). Since *Emx2* transcripts are detected in the lateral utricle three days earlier than Lat HC formation (*Figure 2—figure supplement 5*; *Jiang et al., 2017*), we reasoned that if endogenous Emx2 mediates hair bundle reversal via patterning or cell-fate change, effects of ectopic *Emx2* may be similar and could take days to reverse hair bundle orientation. We crossed two different strains of *cre* mice, *Atoh1$^{Cre}$* or *Gfi1$^{Cre}$* to *Rosa$^{Emx2}$* mice, which resulted in some offspring showing specific expression of *Emx2* in all the HCs. Both Atoh1 and Gfi1 are transcription factors important for HC formation (*Bermingham et al., 1999*; *Wallis et al., 2003*), and lack of *Atoh1* or *Gfi1* results in loss of HCs. Atoh1 is the earliest known transcription factor that commits HC fate in the inner ear (*Bermingham et al., 1999*; *Zheng and Gao, 2000*). However, *Atoh1* expression is not affected in *Gfi1* knockout mice, suggesting that Gfi1 is required later than Atoh1 during HC differentiation

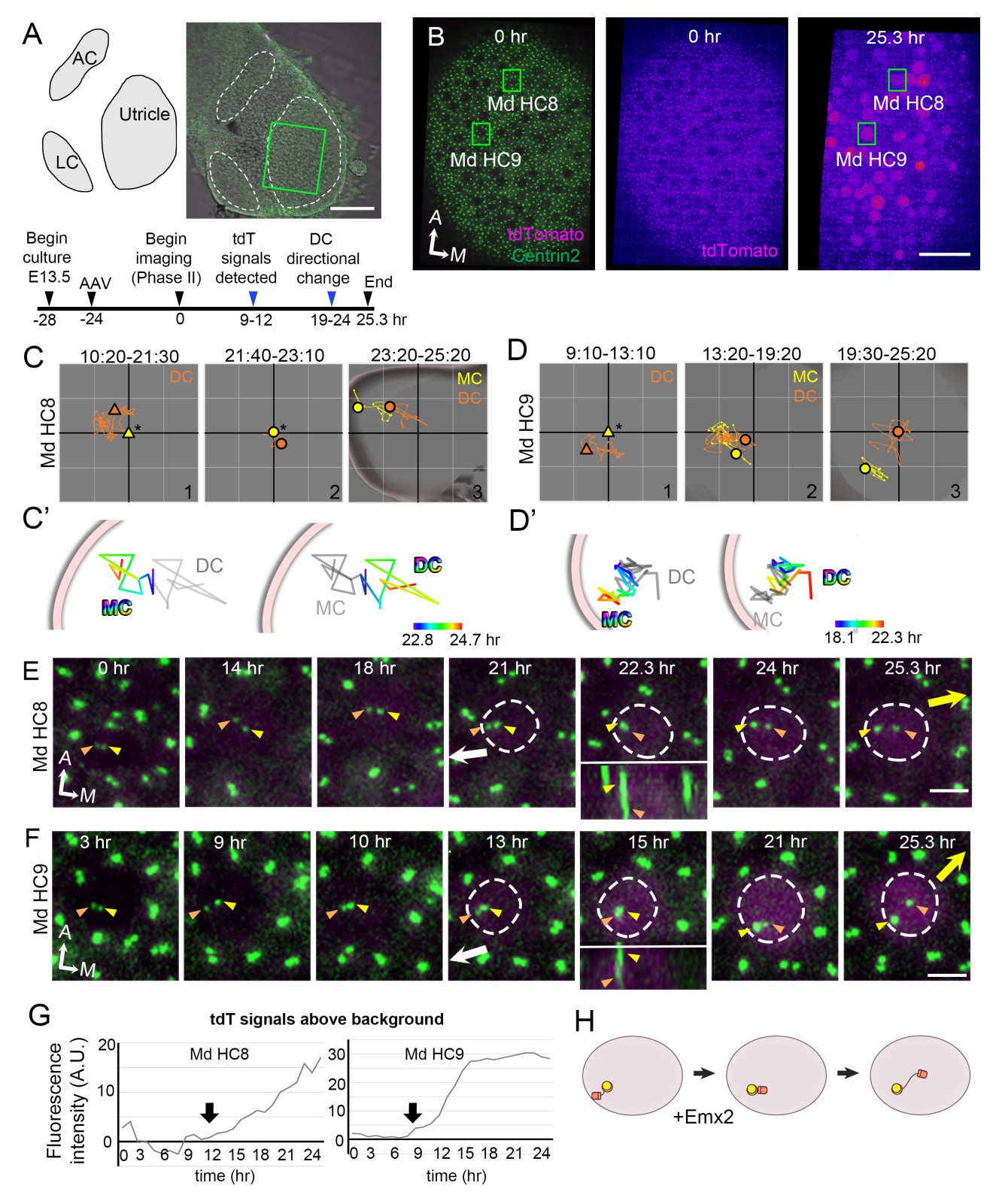

**Figure 5.** Altered daughter centriole (DC) trajectory in medial hair cells (Md HCs) infected with *AAV-Emx2*. (**A**) Schematic and low-magnification image of the *CAG:GFP-Centrin2* utricular explant infected with AAV-Emx2-tdT at E13.5. The timeline of experimental treatments (black arrowheads) and observations (blue arrowheads) are shown. (**B**) An utricular explant at the beginning (0 hr) and end (25.3 hr) of a time-lapse recording showing clear tdTomato-positive cells by the end of recording. (**C, C', D, D'**) Total (**C, D**) and selected temporal (**C', D'**) trajectories of the mother centriole (MC) and

*Figure 5 continued on next page*

Figure 5 continued

DC in Md HC8 and Md HC9 (*Figure 5—source data 1*). Initially, the DC (orange triangle, #1) is positioned lateral to the MC (yellow triangle) at the center (asterisk). Then, the DC (orange dot) moves sporadically around MC (yellow dot, #2), followed by DC moving medial to the MC in #3. (E, F) Selected frames of the recording of Md HC8 and Md HC9. At the beginning, the DC is located by the lateral side (white arrow) of each HC. Then, the DC overlaps with the MC briefly at 22.3 hr for Md HC8 and 15 hr for Md HC9 during recording (insets showing side views), followed by the DC moving medial to the MC toward the medial direction (yellow arrow). (G) tdTomato expression compared to the background level, indicating tdTomato signals exceeded background after 12 (Md HC8) and 9 hr (Md HC9) of recordings (arrows). (H) Schematic of centriole movements in the presence of Emx2. Scale bars: 100 µm in (A), 30 µm in (B) and 3 µm in (E) and (F).

The online version of this article includes the following video and source data for figure 5:

**Source data 1.** Coordinates of centriole positions relative to the center of the hair cell (HC) or the mother centrioles (MCs) in Md HC8 and Md HC9.
**Figure 5—video 1.** Md HC8 and Md HC9.
https://elifesciences.org/articles/59282#fig5video1

(*Wallis et al., 2003*). Thus, the induction of *Emx2* using *Gfi1^Cre^* is expected to be later than that of *Atoh1^Cre^* in the HC lineage.

We first live-imaged *Atoh1^Cre^; Rosa ^Emx2/tdT^; CAG:GFP-Centrin2* utricles (*Figure 3D–F*), in which all Md HCs showed opposite hair bundle orientation from controls by E15.5 (*Figure 3—figure supplement 1*). Live-imaging results showed that the trajectory of centrioles in Md HCs ectopically expressing *Emx2* showed the DC moving around the MC sporadically (*Figure 3E*#1, 3F#1–4), similar to normal Md HCs at Phase I (*Figure 3B*#1, 3C#1–3). Then, the DC migrated toward the medial periphery (*Figure 3E*#2, 3E', 3F#5, yellow arrow, *Figure 3—source data 1*, *Figure 3—video 1*), opposite from the normal lateral direction (*Figure 3F*, white arrow) and controls (*Figure 3B–C*). This trajectory was followed by the MC (*Figure 3E*#3, 3F#6–7). This pattern of centriole migration from center of the HC to the medial edge occurred in 60% of the HCs analyzed (*Figure 3J*, *Figure 3—source data 2*). The remaining HCs exhibited a pattern that is similar to the *Gfi1^Cre^; Rosa^Emx2/tdT^* Md HCs described below, suggesting that there are two modes of centriole trajectory.

In *Gfi1^Cre^; Rosa^Emx2/tdT^* utricles (*Figure 3G–I*), in which all the hair bundle orientation in Md HCs are known to be reversed (*Jiang et al., 2017*), live imaging results showed that some of the centriole trajectories were different from those in *Atoh1^Cre^; Rosa^Emx2/tdT^* utricles (*Figure 3G–J*, *Figure 3—figure supplement 2, A–C*, *Figure 3—source data 2*, p=0.0010). At the beginning of the recording, *Gfi1^Cre^; Rosa^Emx2/tdT^* Md HCs analyzed already showed tdTomato expression and the DC was asymmetrically located towards the lateral side (*Figure 3H*#1, 3I#1, Md HC4, *Figure 3—figure supplement 2C–C''*#1, Md HC5-7, white arrow, *Figure 3—video 1*), resembling Md HCs at Phase II (*Figure 3C*). Within 3 hr (*Figure 3—figure supplement 2, C'*#1–3, Md HC6, *Figure 3—figure supplement 2—source data 1*) to 9 hr (*Figure 3I*#1–4, Md HC4, *Figure 3—figure supplement 2, C''*#1–4 Md HC7) of recordings, the DC remained lateral to the MC. Thereafter, the distance between the DC and MC was reduced and sometimes the two centrioles were transiently aligned along the apical-basal axis (*Figure 3I*#5 Md HC4, *Figure 3—figure supplement 2, C*#5–6 Md HC5, C'#5–6 Md HC7, *Figure 3—figure supplement 2—source data 1*). Then, the DC switched position to the medial side of MC (*Figure 3H*#3, 3H', 3I#6–7, *Figure 3—figure supplement 2, C–C''*, yellow arrow). These results suggest that in *Gfi1^Cre^; Rosa^Emx2/tdT^* Md HCs, the centrioles are initially aligned in the normal lateral positions primed for hair bundle establishment but upon activation of *Emx2* transcription mediated by *Gfi1*-driven Cre, the two centrioles reverse their positions to the medial side. This lateral to medial centriole reversal pattern was also observed in *Atoh1^Cre^; Rosa^Emx2/tdT^* utricles but at a lower frequency than Md HCs of *Gfi1^Cre^; Rosa^Emx2/tdT^* (*Figure 3J*, 40% vs 84.6%, p=0.0010, *Figure 3—source data 2*). These frequency differences in migration patterns between the two *cre* lines are consistent with the notion that *Emx2* activation using *Gfi1^Cre^* is later than *Atoh1^Cre^*, thus a higher percentage of centrioles in *Gfi1^Cre^* samples were observed to migrate from the destined lateral periphery toward the medial rather than directly from the central toward the medial position (*Figure 3J*, 60% vs 15.4%). Furthermore, our results indicate that centriole trajectory during both Phase I and Phase II is plastic and can be altered by Emx2.

## Infections with AAV-Emx2 reverse hair bundle orientation

Our time-lapse recordings showed that the longest time observed for *Gfi1^Cre^; Rosa^Emx2/tdT^* Md HCs to reverse centriole positions from the lateral to medial periphery was approximately 12 hr (*Figure 3*,

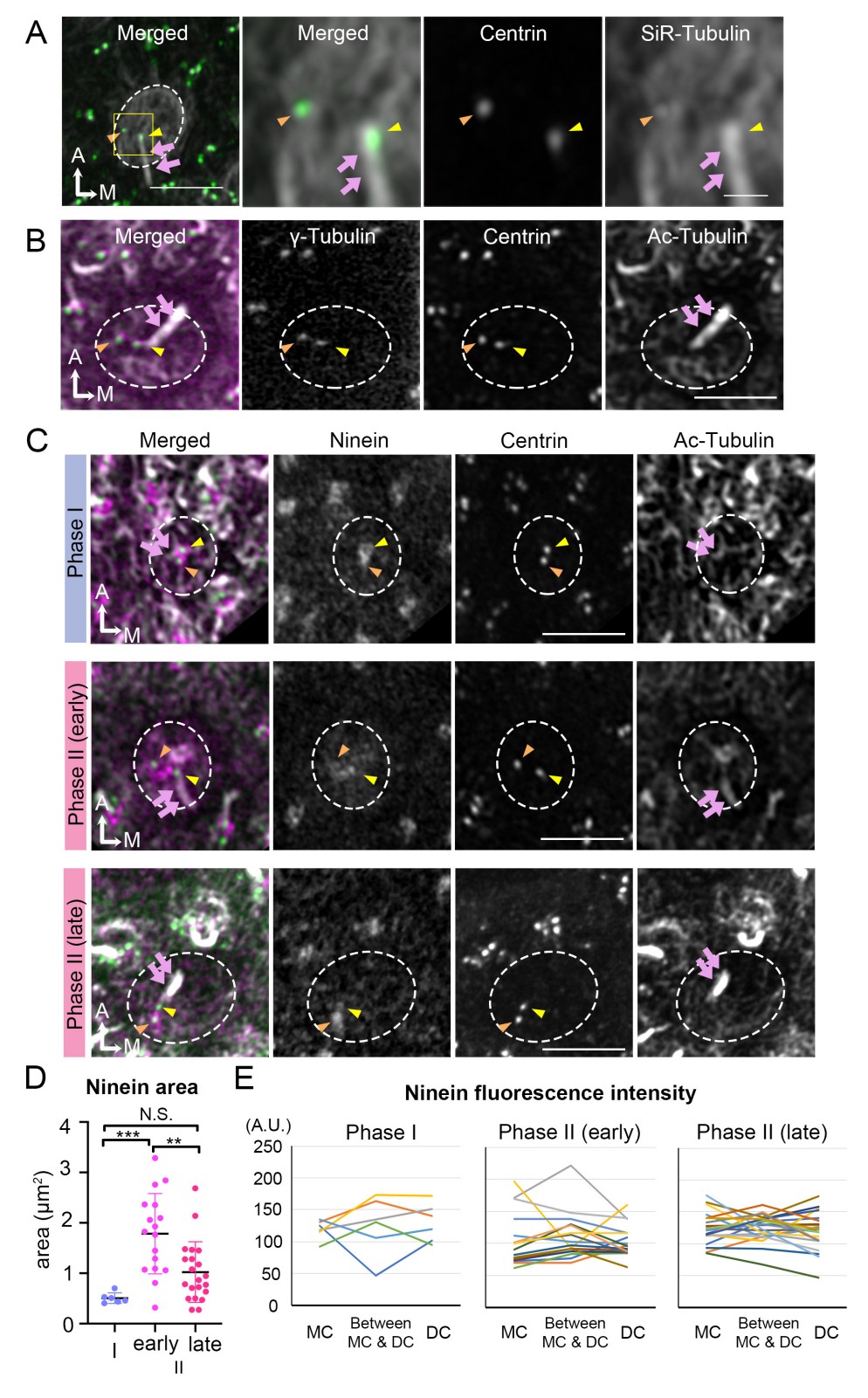

**Figure 6.** Broaden ninein localization during centriole migration. (**A**) SiR-tubulin (white) labeling of a medial hair cell (Md HC) at Phase II from an
*Atoh1^Cre; Rosa^tdT/+; CAG:GFP-Centrin2* utricule at E13.5. The magnified views of the yellow rectangular area in the left panel are shown in the three
right panels, illustrating that both centrioles (green in the merged picture) are associated with the microtubule network. MC, yellow arrowhead; DC,
orange arrowhead; kinocilium, pink arrows. Dotted white lines indicate apical surface of the HC. (**B**) Anti-γ-tubulin staining of a Md HC at Phase II.
*Figure 6 continued on next page*

*Figure 6 continued*

Daughter (DC) and mother centrioles (MC) show similar expression of γ-tubulin (magenta in the merged picture). Anti-acetylated tubulin antibody labels mature microtubules and the kinocilium. (**C**) Immunostaining of ninein in Md HCs in Phase I, early and late Phase II. Phase I and late Phase II HCs show centrosomal ninein (magenta on merged picture) staining. At the beginning of Phase II, ninein staining is diffuse and broader than the centrioles. (**D**) Distribution of the ninein-positive area (μm$^2$) during Phase I, early and late Phase II HCs. (**E**) Ninein fluorescence intensity associated with the MC, DC and the region in-between the two centrioles during Phase I, early and late Phase II HCs (see Materials and methods). No consistency of ninein staining associated with a specific centriole or region was observed among various samples (numbers of HCs for Phase I, Phase II early and late are 6, 17, and 21, respectively). Scale bars: bar in left panel of (**A**) equals 3 μm, bar in the right panel of (**A**) equals 1 μm and applies to the two adjacent panels on its left, bars on the fourth and third panels of respective (**B**) and (**C**) equal 3 μm and apply to other panels in (**B–C**). **p<0.01, ***p<0.001.

The online version of this article includes the following source data for figure 6:

**Source data 1.** Quantification of ninein area in hair cells (HCs) during each phase of centriole migration.
**Source data 2.** Anti-ninein staining intensity associated with the mother centriole (MC), daughter centriole (DC) and area in between the centrioles.

---

Md HC4, *Figure 3—figure supplement 2*, Md HC7) suggesting that Emx2 could exert its effects on hair bundle reversal within hours. However, the precise time frame between the onset of *Emx2* expression driven by *Gfi1*-Cre and reversal of centriole positions remains unclear. To further investigate the time required by Emx2 to alter hair bundle orientation, we ectopically expressed *Emx2* in utricular explants using AAV2.7m8 adeno-associated virus, which has been shown to infect cochlear HCs efficiently (*Isgrig et al., 2019*). We infected CAG:*GFP-Centrin2* utricular explants at E13.5 with an AAV2.7m8 adeno-associated viral vector, AAV2.7m8-CAG-Emx2-P2A-tdTomato (AAV-Emx2-tdT), in which both *Emx2* and *tdTomato* transcripts were driven under the universal CAG promoter. Forty-eight hr after infection, approximately 70% of total HCs were infected (*Figure 4C*, 71.1 ± 6.3% with AAV-Emx2-tdT, *Figure 4—source data 1*), which was similar to AAV-tdT controls (72.8 ± 5.6%). The void of anti-β2-spectrin staining indicates the kinocilium position in the apical HC surface and reveals the hair bundle orientation (*Deans et al., 2007*). In AAV-tdT controls, infected HCs showed normal hair bundle orientation (*Figure 4A*, white arrows, 4C). However, 13.9 ± 7.2% of HCs infected with AAV-Emx2-tdT showed opposite kinocilium position (*Figure 4B* yellow arrows, 4C) versus none in controls, indicating that AAV-Emx2-tdT is sufficient to reverse hair bundle orientation. The efficiency of changing kinocilium position was not correlated with the expression levels of ectopic Emx2 (*Figure 4E*, *Figure 4—source data 2*), as we found high Emx2 expressors with normal bundle orientation (*Figure 4D and D'*, white arrow with double arrowheads). At the same time, although high Emx2 expressors were found with opposite bundle orientation (yellow arrow with asterisk) from the normal orientation (white arrows), low Emx2 expressors could also change bundle orientation equally well (*Figure 4D'*, yellow arrow with arrowhead).

Next, we live-imaged the centriole reversal process in AAV-Emx2-tdT infected CAG:*GFP-Centrin2* utricular cultures (*Figure 5A*). At 24 hr after viral infection, the majority of the cells in the utricular explants were tdTomato-negative (*Figure 5B*) but many HCs turned on tdTomato by 36 hr after infection (*Figure 5E and F*). At the beginning of recordings, most of the infected Md HCs showed the DC lateral to the MC, even though the outline of the HCs was not evident yet due to the lack of tdTomato signal (*Figure 5C*#1, D#1, E, F, *Figure 5—video 1*). As tdTomato expression became apparent, the peripheral location of the two centrioles was confirmed, indicating that these cells were at late Phase II (*Figure 5E–F*, Md HC8, 21 hr, Md HC9, 13 hr). Then, the two centrioles moved to align longitudinally with each other transiently (*Figure 5E–F*, small panel insets, 22.3 hr for Md HC8 and 15 hr for Md HC9), followed by the DC moving medial to the MC (*Figure 5C*#3, 5C', 5E 24–25.3 hr, Md HC8, 5D#3, 5D', 5F 21–25.3 hr, Md HC9), suggesting a change in the course of DC trajectory from lateral to medial periphery. Quantification of tdTomato signal that was above background in infected cells showed that tdTomato signals were elevated by 9–12 hr of imaging (*Figure 5A,G*, 1.5 days after infection), although the signals may not be apparent in the single time frame images (*Figure 5E,F*). Then, DC positional reversal occurred within 10–12 hr after the detectable tdTomato signals. Using positive tdTomato signals as a proxy for Emx2 activation, these results suggest that Emx2 can reverse centriole trajectory within 10–12 hr. Thus, both the viral approach using AAV and genetic approach using *Gfi1$^{Cre}$; Rosa$^{Emx2/tdT}$* utricles suggest that Emx2 is able to mediate hair bundle reversal within a short period of time (*Figure 5H*) despite onset of endogenous *Emx2* expression occurring 3 days prior to HC formation (*Figure 2—figure supplement 5*).

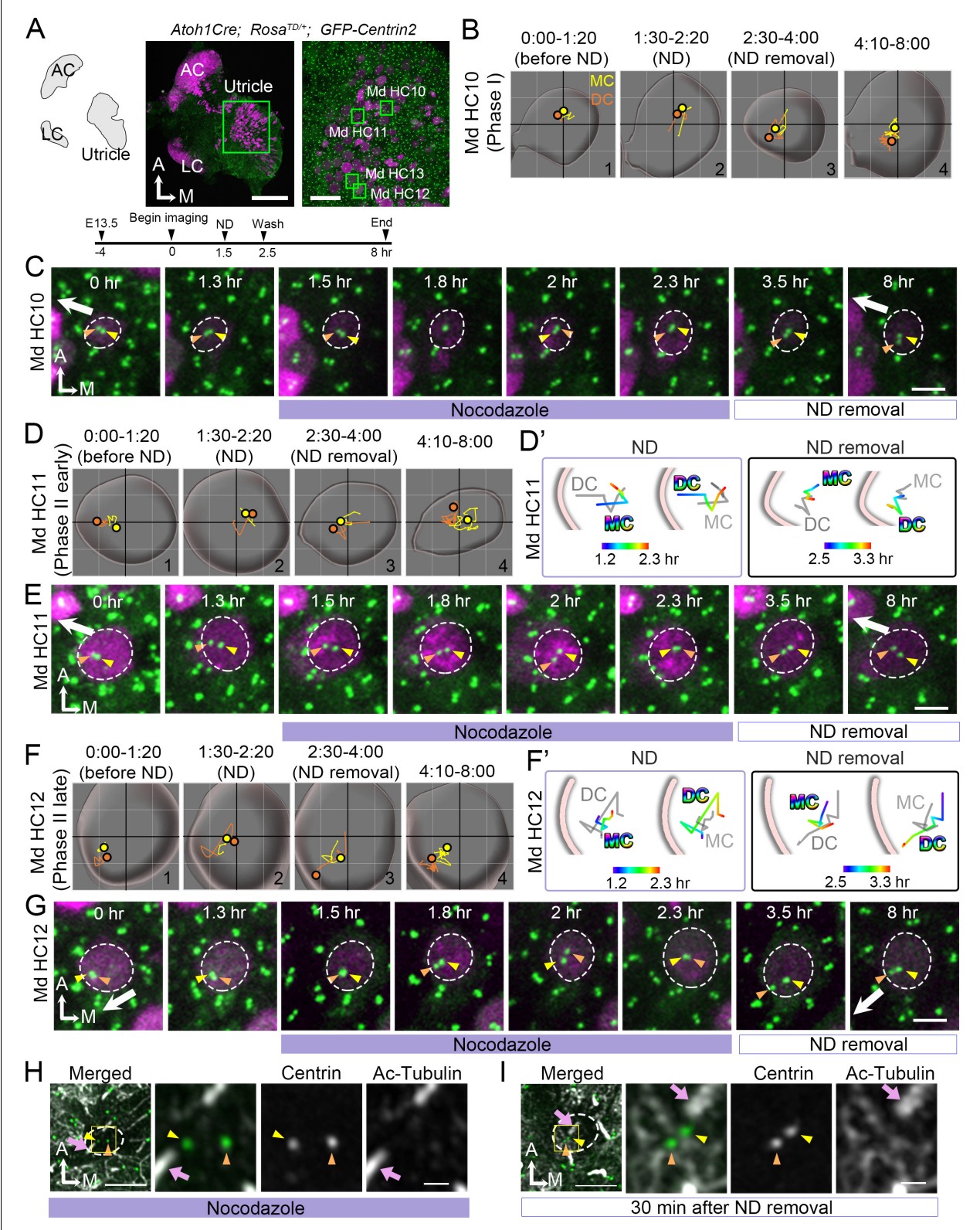

**Figure 7.** Nocodazole treatments affect centriole migration and positions. (**A**) Schematic, low and high magnifications of an *Atoh1^Cre^; Rosa^tdT/+^; CAG: GFP-Centrin2* utricular explant at E13.5. The timeline of the experiment is shown. (**B, C**) Md HC10 at Phase I when centrioles are at the center of the hair cell (HC). (**B**) Total trajectories as well as (**C**) selected frames of mother centriole (MC) (yellow) and daughter centriole (DC) (orange) in Md HC10 (*Figure 7—source data 1*). MC and DC remained in the center during nocodazole treatment (B#2, **C**). After nocodazole removal, the DC started to

*Figure 7 continued on next page*

*Figure 7 continued*

move to the periphery (B#3,4, C, n = 3 out of 6 Phase I HCs). (**D–E**) Md HC11 at the beginning of Phase II, in which the DC is located peripherally whereas the MC is at the center. (**D**) Total, selected temporal (**D'**) trajectories as well as selected frames (**E**) of MC and DC in Md HC11 (*Figure 7—source data 1*). During nocodazole treatment, both DC and MC relocated to the center of the HC (D#2, **D'**). After drug removal, the DC moved toward the periphery within 1 hr (**D'**, **E**, 3.5 hr, n = 11 out of 12 early Phase II HCs). (**F–G**) Md HC12 at late Phase II in which both centrioles are peripherally positioned. (**F**) Total, selected temporal (**F'**) trajectories as well as selected frames (**G**) of MC and DC in Md HC12 (*Figure 7—source data 1*). During nocodazole treatment, both centrioles move back to the center of the HC (F#2, **F'**, **G**). After removal of nocodazole, centrioles return to the periphery with the DC moving ahead of the MC (**F'**, n = 17 out of 18 late Phase II HCs). (**H**) The nocodazole-treated Md HC shows mispositioned centrioles (MC, yellow arrowhead; DC, orange arrowhead) that are no longer asymmetrically located in the periphery. The three panels on the right are merged, single centrin and acetylated tubulin images of the rectangular area in the left panel. Tubulin arrays are absent in the cytoplasm of HC. (**I**) A Md HC after removal of nocodazole for 30 min shows centrioles returning to their peripheral location. The three panels on the right are magnifications of the rectangular area in the left panel. Tubulin arrays (white) are radiating from the centrioles to the periphery Scale bars: 100 μm (low magnification) and 30 μm (high magnification) in (**A**), 3 μm in (**C**), (**E**), (**G**), and the first panel in (**H**) and (**I**), and 1 μm in high magnification of (**H**) and (**I**).

The online version of this article includes the following video, source data, and figure supplement(s) for figure 7:

**Source data 1.** Coordinates of centriole positions relative to the center of the cell for Md HC10, MD HC11 and Md HC12.
**Figure supplement 1.** A late Phase II HC treated with nocodazole.
**Figure supplement 1—source data 1.** Coordinates of centriole positions relative to the center of Md HC13.
**Figure 7—video 1.** MD HC12 and MD HC13 - nocodazole treatment and removal.
https://elifesciences.org/articles/59282#fig7video1

## Microtubules are required to stabilize the asymmetrical location of the centrioles

Thus far, our live-imaging results indicate that the migration of the DC always preceded that of the MC under either control or treated conditions. To identify potential qualitative differences between the two centrioles that could account for the migration pattern, we investigated their association with microtubules and proteins related to microtubule nucleation and anchoring. Using SiR-tubulin, a cell permeable fluorogenic probe for microtubules, we labeled microtubules in live *Atoh1^Cre^; Rosa^tdT/+^; CAG:GFP-Centrin2* utricular cultures. We showed that both centrioles were associated with microtubules in tdTomato-positive HCs determined to be Phase II, based on the position of the centrioles (*Figure 6A*). Immunostaining with anti-γ-tubulin antibodies indicated that both centrioles in Phase II HCs are associated with microtubule nucleation (*Figure 6B*). Ninein is a centrosomal protein that has both microtubule nucleation and anchoring functions (*Delgehyr et al., 2005*) and it is preferentially associated with the MC than the DC in somatic cells, serving its microtubule nucleation role (*Betleja et al., 2018*; *Piel et al., 2000*). In Phase I HCs, in which centrioles are centrally located, ninein staining was concentrated by the two centrioles (*Figure 6C*, Phase I, n = 6). As DC started to migrate towards the periphery and the two centrioles became further apart, ninein staining became broadly distributed surrounding both centrioles (*Figure 6C*, early Phase II, n = 17). However, ninein staining was concentrated at the centrioles again at late Phase II (*Figure 6C*, n = 21). Even though the association of ninein with specific centrioles or the location in between the two centrioles is inconsistent and does not distinguish between the two centrioles (*Figure 6E*, *Figure 6—source data 2*), the broad distribution of ninein staining beyond the centrioles during centriole migration is significantly different from other stages of the trajectory (*Figure 6D*, Phase I vs early Phase II, p=0.0005; early Phase II vs late Phase II, p=0.0025; Phase I vs late Phase II, p=0.21, *Figure 6—source data 1*). This broad distribution of ninein staining beyond the centrioles during centriole migration suggests that ninein may serve as microtubule anchoring during this period.

Previous reports in cochlear explants proposed that microtubule plus ends attached to the LGN/Insc/Gαi complex relocate the MC/kinocilium through microtubule shortening and/or dynein mediated mechanism to the periphery (*Ezan et al., 2013*). Blocking Gαi with pertussis toxin disrupted the microtubule plus-end binding protein, EB1 and the kinocilium positioning. We tested this hypothesis in our live utricular culture and asked whether the DC behaves in a similar manner as MC. We treated utricular explants with nocodazole, which disrupts the microtubules by binding to free tubulin dimers and inhibits microtubule polymerization (*Hoebeke et al., 1976*) and we analyzed nocodazole effects on all stages of centriole migration (*Figure 7*). Nocodazole was introduced for 1 hr at 1.5 hr into the live-imaging and recording continued for 5.5 hr after drug removal (*Figure 7A*). In Phase I HCs, in which centrioles were centrally located at the beginning of the recording

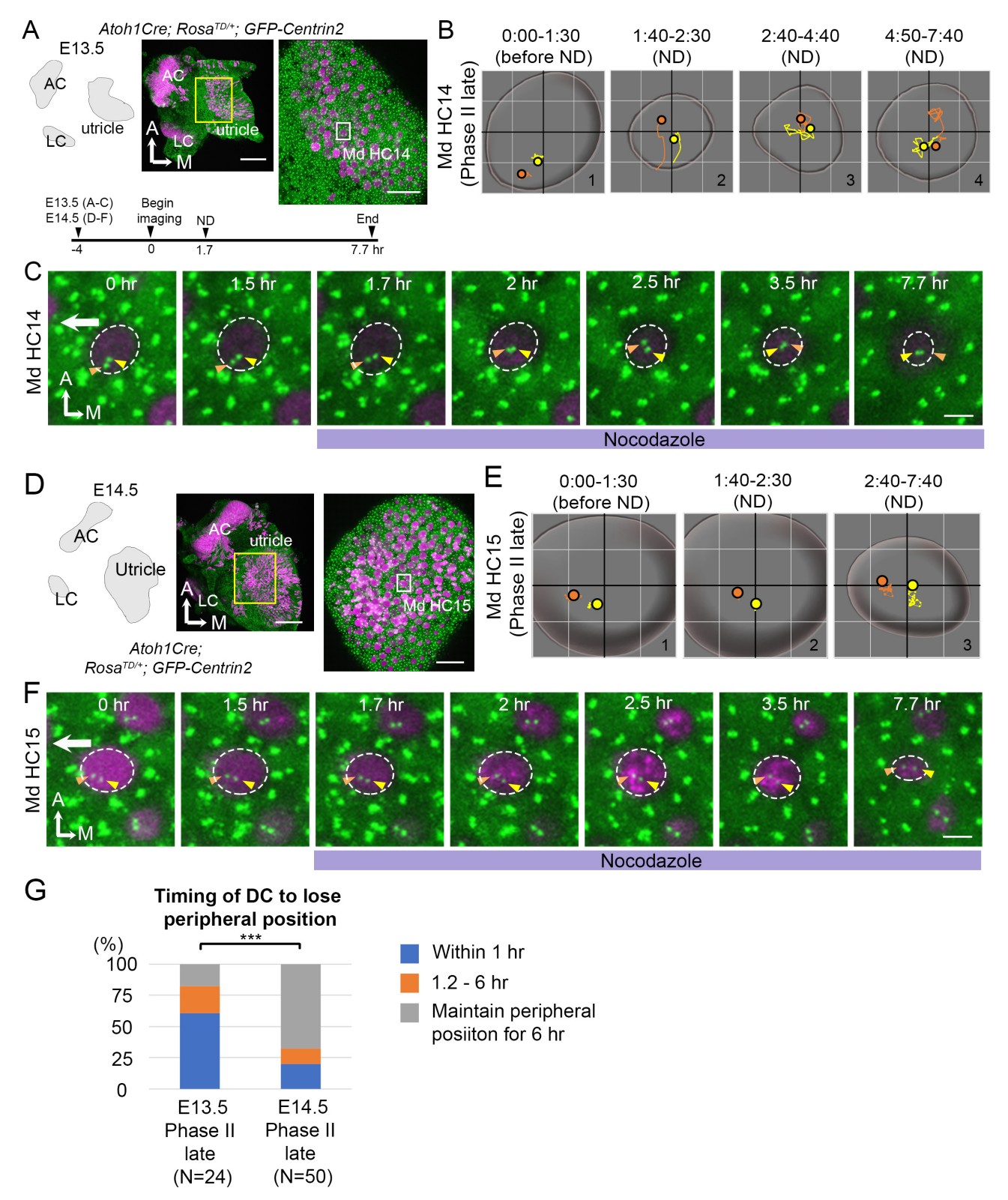

**Figure 8.** Mature HCs are less sensitive to the microtubule disruption. (**A**) Schematic, low and high magnifications of an *Atoh1^Cre^; Rosa^tdT/+^; CAG:GFP-Centrin2* utricular explant at E13.5. The timeline of the experiment is shown. (**B**) Total trajectories as well as (**C**) selected frames of MC (yellow) and DC (orange) in Md HC14 at late Phase II (*Figure 8—source data 1*). During nocodazole treatment, the centrioles move back to the center of the HC within 30 min (B#2, C 2 hr). Then, both centrioles moved around in the center of the HC in variable relative positions from each other till the end of the

*Figure 8 continued on next page*

*Figure 8 continued*

experiment (B#2–4, C 2.5–7.7 hr). (**D**) Schematic, low and high magnifications of an *Atoh1^Cre^; Rosa^tdT/+^; CAG:GFP-Centrin2* utricular explant at E14.5. (**E–F**) Total trajectories (**E**) as well as selected frames (**F**) of MC and DC in Md HC15 at late Phase II (*Figure 8—source data 1*). The peripheral location of the centrioles and the relative relationship of DC lateral to the MC were maintained until the end of recording (E#2-#3, F 1.5–7.7 hr). The surface of the HC appeared reduced in size by the end of the experiment (E and F, 7.7 hr). (**G**) Timing of DC to lose the peripheral position. At E13.5, 60.7% (17 out of 28 HCs) of late Phase II HCs lose their peripheral positions of DCs within 1 hr-treatment of ND, whereas, at E14.5, it reduces to 20% (10 out of 50, p=0.000089, *Figure 8—source data 2*). Scale bars: 100 μm (low magnification) and 30 μm (high magnification) in (**A**) and (**D**), 3 μm in (**C**) and (**F**). \*\*\*p<0.001.

The online version of this article includes the following source data for figure 8:

**Source data 1.** Coordinates of centriole positions relative to the center of the cell for Md HC14 and MD HC15.

**Source data 2.** Timing of the daughter centriole (DC) to lose its peripheral position under nocodazole treatments.

(*Figure 7B*#1, 7C 0–1.3 hr), centrioles remained in the apical center of the HC even though the DC continued to move around the MC during nocodazole treatments (*Figure 7B*#2, 7C 1.5–2.3 hr). After drug removal, some of the HCs showed migration of DC towards the periphery (*Figure 7B*#3-#4, 7C 3.5–8 hr, n = 3 out of 6 HCs analyzed). The rest of the Phase I HCs remained in Phase I for 5.5 hr after nocodazole removal (not shown). In early Phase II HCs where there was a bigger distance between the centrioles and the DC was positioned closer to the periphery (*Figure 7D*#1, 7E 0–1.3 hr), nocodazole treatments caused the DC to move centrally (*Figure 7E* 1.5–2.3 hr, 7D#2, 7D'). The DC continued to move sporadically around the MC (*Figure 7D*#2, 7E 1.5–2.3 hr), resembling the behavior observed in Phase I of nascent HCs (*Figure 2*). After nocodazole removal, the DC started to migrate towards the periphery again ahead of the MC (*Figure 7D*#4, D', 7E 3.5–8 hr, n = 11 out of 12 HC analyzed).

HCs at the late Phase II show both centrioles located in the periphery (*Figure 7F*#1, 7G 0–1.3 hr). Similar to the phenomenon observed with HCs at the beginning of Phase II (*Figure 7D–E*), both centrioles returned to the center of the HC in the presence of nocodazole within 30 min (*Figure 7F*#2, 7F', 7G 1.5–2 hr, *Figure 7—source data 1*, *Figure 7—figure supplement 1*, *Figure 7—figure supplement 1—source data 1*), and the DC continued to move sporadically around the relatively stable MC (*Figure 7F*#2, 7G 2–2.3 hr). Once nocodazole was washed out, two centrioles returned to the lateral periphery within a few hr with the DC moving ahead of the MC (*Figure 7F*#3-#4, 7F', 7G, 3.5–8 hr, *Figure 7—video 1*, n = 17 out of 18 HCs analyzed). These results indicate that both the DC and MC respond to nocodazole treatments in a similar manner. Furthermore, acetylated-tubulin staining of HCs, which labels stable microtubules, showed a loss of tubulin arrays along with mislocalized DC and MC from the periphery during nocodazole treatment (*Figure 7H*). After nocodazole removal, tubulin arrays were re-established and both centrioles recovered their positions in the periphery (*Figure 7I*). Together, these results suggest that both the DC and MC are dependent on an active force that translocate them to the periphery via an intact microtubule network.

## Centriole positions after establishment are less sensitive to microtubule disruption

Nocodazole treatments affected centriole migration at E13.5. However, we observed some HCs with peripherally located centrioles (late Phase II) that were unaffected by 1 hr of nocodazole treatments (n = 8 out of 30 HCs analyzed). It was not clear whether these HCs required longer than 1 hr of exposure to nocodazole in order to respond or they represented HCs that were more mature and thus insensitive to microtubule disruption. To distinguish these possibilities, we treated E13.5 and E14.5 utricle explants with nocodazole for 6 hr and followed centriole trajectories in HCs (*Figure 8*). We reasoned that if the non-responsive HCs at E13.5 represented more mature HCs, then this population of HCs should increase in older utricles at E14.5. We observed that 6 hr of nocodazole treatments at E13.5 resulted in rapid central migration of centrioles within an hr (*Figure 8B*#2, 8C 1.7–2 hr, 8G), similar to treatments for 1 hr (*Figure 7*), and the centrioles stayed in the center with the DC in variable positions relative to the MC (*Figure 8B*#2-#4, 8C 2–7.7 hr). However, a majority of the HCs at E14.5 did not respond to nocodazole treatments in a similar manner. Only 20% of the HCs analyzed showed their centrioles moved away from the periphery within the first hr, compared to 60.3% of the HCs at E13.5 (*Figure 8G*, p=0.000089, *Figure 8—source data 2*) and the relative

positions between the two centrioles always maintained with the DC being more lateral than the MC (*Figure 8E*#2-#3, 8F 1.7–3.5 hr). By the end of the experiment, although the two centrioles seemed more centrally displaced, the apical surface of the HC was also reduced (*Figure 8E*#3, 8F 7.7 hr). It is not clear whether the central migration of the centrioles by the end of the experiment was secondary due to reduction of the apical HC surface. Although the apical surface of HCs at E13.5 also appeared smaller by 6 hr of nocodazole treatments (*Figure 8C* 7.7 hr), the response of centrioles to nocodazole was much more acute and profound at E13.5 than E14.5. Together, these results suggest that as HCs mature, they are less sensitive to microtubule disruption.

## Correlation of centriole trajectory with Gαi and cuticular plate formation

Thus far, our results showed no difference between the DC and MC in their association with the microtubule network and nucleation center as well as their response and recovery from nocodazole treatments. Lastly, we investigated the association of the polarity complex, LGN/Insc/Gαi, with centriole migration. We correlated the timing of centriole trajectory with the formation of the LGN/Insc/Gαi complex using anti-Gαi antibody staining as well as formation of the cuticular plate using anti-spectrin antibody. Our results showed that during Phase I when both centrioles were at the center of the HC, Gαi staining was diffused on the apical HC surface whereas β2-spectrin staining was not detectable (*Figure 9A and A'*). By the beginning of Phase II, when the DC was moving towards the periphery whereas the MC remained relatively closer to the center of the HC, Gαi staining was concentrated at the lateral side of the HC where the future hair bundle will be established (*Figure 9B and B'*). The DC being more laterally positioned than the MC was better associated with the Gαi domain than the MC (*Figure 9B'*). By this stage, β2-spectrin staining was also apparent, located in the medial side of the HC (*Figure 9B and B'*). This complementary expression patterns between Gαi and β2-spectrin became more strengthened and evident by the time centrioles reached late Phase II (*Figure 9C and C'*). Although blocking Gαi activity using pertussis toxin primarily affects hair bundle orientation in the lateral and not the medial utricle (*Figure 9—figure supplement 1*; *Jiang et al., 2017*), our centriole tracking and immunostaining results suggest that the polarity complex LGN/Insc/Gαi is established at the same time that centrioles started to migrate and these processes precede cuticular plate formation.

## Discussion

### Hair bundle establishment in HCs

Our live-imaging study is the first extensive time-lapse imaging of hair bundle acquisition in mammalian HCs. Imaging results of centriole migration in utricular HCs are consistent with a previous model extrapolated from results of SEM that the kinocilium starts out in the center of a HC before reaching its peripheral destination for hair bundle establishment (*Dabdoub et al., 2003*; *Lu and Sipe, 2016*; *Denman-Johnson and Forge, 1999*). Based on results in the chicken basilar papilla, it was proposed that the kinocilium undergoes fairly extensive migration along the periphery of the HC before reaching its final destination (*Figure 1A*; *Cotanche and Corwin, 1991*; *Tilney et al., 1992*). However, more recent studies in the mouse cochlea suggest that the MC/kinocilium takes a more direct route from the center of the HC to its final destination in the periphery (*Dabdoub et al., 2003*; *Lu and Sipe, 2016*; *Denman-Johnson and Forge, 1999*; *Montcouquiol et al., 2003*). Additionally, a similar time-lapse recording of embryonic cochlear explants of approximately 3 hr demonstrated a confined Brownian motion of centrioles in HCs (*Lepelletier et al., 2013*). By calculating the mean square distance of centriole movements in the utricular HCs as described (*Lepelletier et al., 2013*), it is likely that centrioles undergo Brownian motion in the utricular HCs but it is not clear whether centriole movements are as confined as in cochlear HCs. Nevertheless, our results are consistent with findings in the mouse that the MC/kinocilium and DC take a direct path to their destination in the periphery (*Figure 10*). Additionally, we found an intriguing centriole migration pattern for hair bundle establishment as discussed below.

It has been proposed that the migration of kinocilium to its destinated location is achieved through an active force on centrioles via the microtubule network. This active force is suggested to be exerted through LGN/Insc/Gαi complex on microtubules by recruiting Lis1/dynein and/or

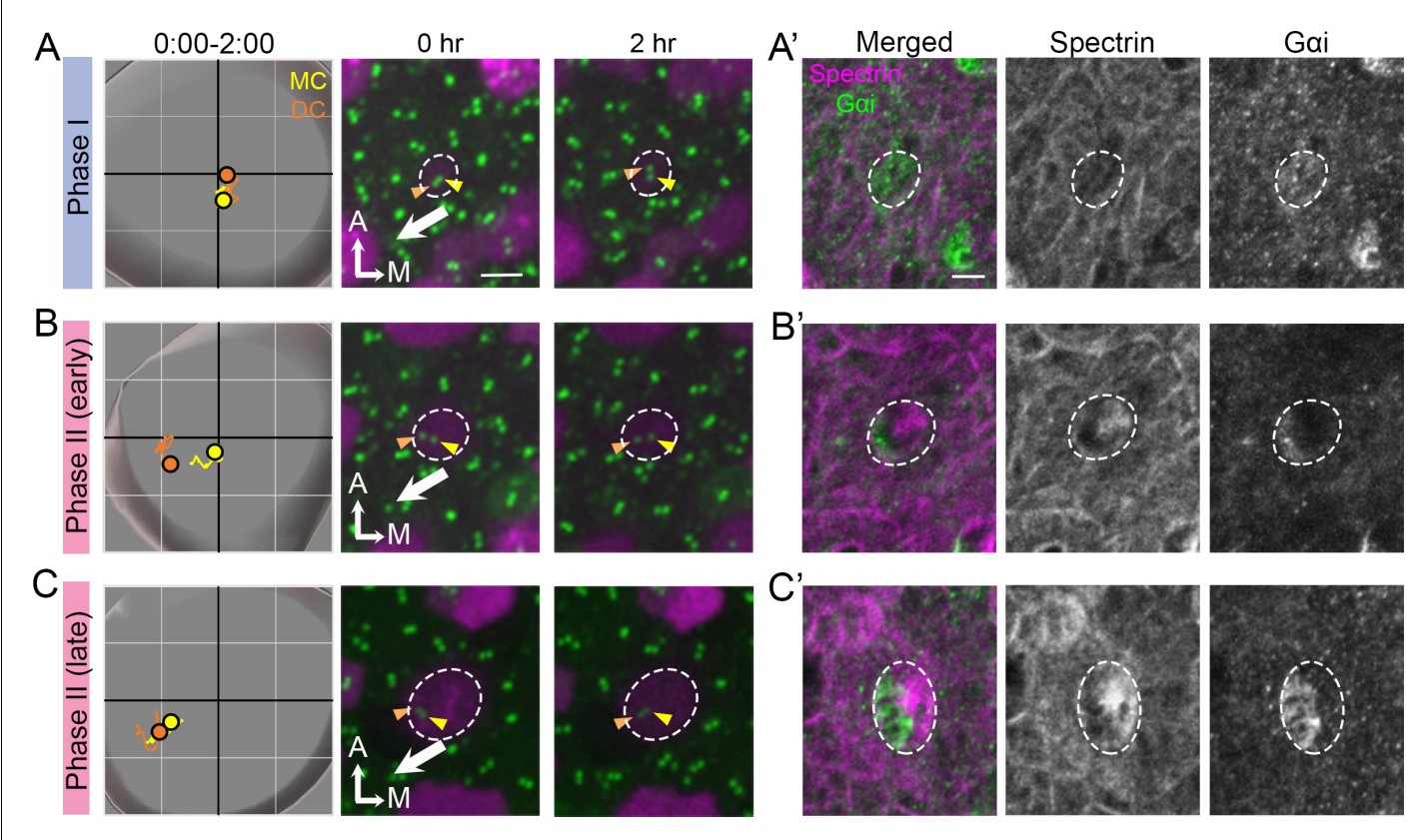

**Figure 9.** Correlation of centriole trajectory with Gαi and β2-spectrin accumulation in nascent HCs. Trajectories and selected frames of typical Phase I (**A**), early (**B**) and late Phase II (**C**) Md HCs during 2 hr live imaging (*Figure 9—source data 1*). (**A'**, **B'**, **C'**) Immunostaining of β2-spectrin and Gαi in the same HCs as shown in **A**, **B** and **C**, after live imaging. (**A'**) At phase I, β2-spectrin signal is not detectable, and Gαi is diffusely expressed in the apical HC surface. (**B'**) At early Phase II, Gαi starts to accumulate in the lateral periphery and β2-spectrin signal is located in the medial side. (**C'**) At late Phase II, both Gαi and β2-spectrin signals become more apparent and complementary to each other. Scale bar of (**A**) equals 3 μm and apply to the other panels in (**A–C**). Scale bar on the first panel of (**A'**) equals 3 μm and apply to the other panels in (**A'–C'**).

The online version of this article includes the following source data and figure supplement(s) for figure 9:

**Source data 1.** Coordinates of centriole positions during the 2 hr of live imaging.

**Figure supplement 1.** Pertussis toxin treatments affect hair bundle orientation in the lateral but not medial utricular hair cells.

shortening of microtubule (*Ezan et al., 2013*; *Lu and Sipe, 2016*; *Tarchini and Lu, 2019*). Disruption of the microtubule network caused the centrioles to promptly return to their initial central location rather than to remain stationary at the periphery of HCs. These results supported this active force hypothesis regardless of the mechanism involved, whether it is generated by pulling or shortening of the microtubules or both. Furthermore, our nocodazole results (*Figure 7*) as well as the anti-Gαi staining (*Figure 9*) indicate that the DC is likely to be under similar control of the LGN/Insc/Gαi complex as the MC. However, once the centrioles acquire their final position and cuticular plate starts to form, centriole positioning becomes less sensitive to microtubule disruption (*Figures 8* and *9*).

The directed migration of the centrioles depends on oriented microtubule arrangement of the minus ends at the centrioles and the plus ends at the periphery. This arrangement raises the possibility that proper anchoring of microtubules within HCs is an important factor. Among the anchoring proteins, we focused on ninein, which has both microtubule nucleation and anchoring functions (*Delgehyr et al., 2005*). It functions as a microtubule nucleation by docking γ-tubulin to the centrosomes and it also anchors microtubules in non-centrosomal site. For example, in pillar cells (supporting cells) of the cochlea, ninein has a non-centrosomal location in adherens junctions, which serve as microtubule anchors (*Mogensen et al., 2000*; *Moss et al., 2007*). Although ninein expression is dynamic and variable during the two phases of centriole migration, the broaden distribution of ninein when centrioles migrate towards periphery of the HC is clear and raises the possibility that

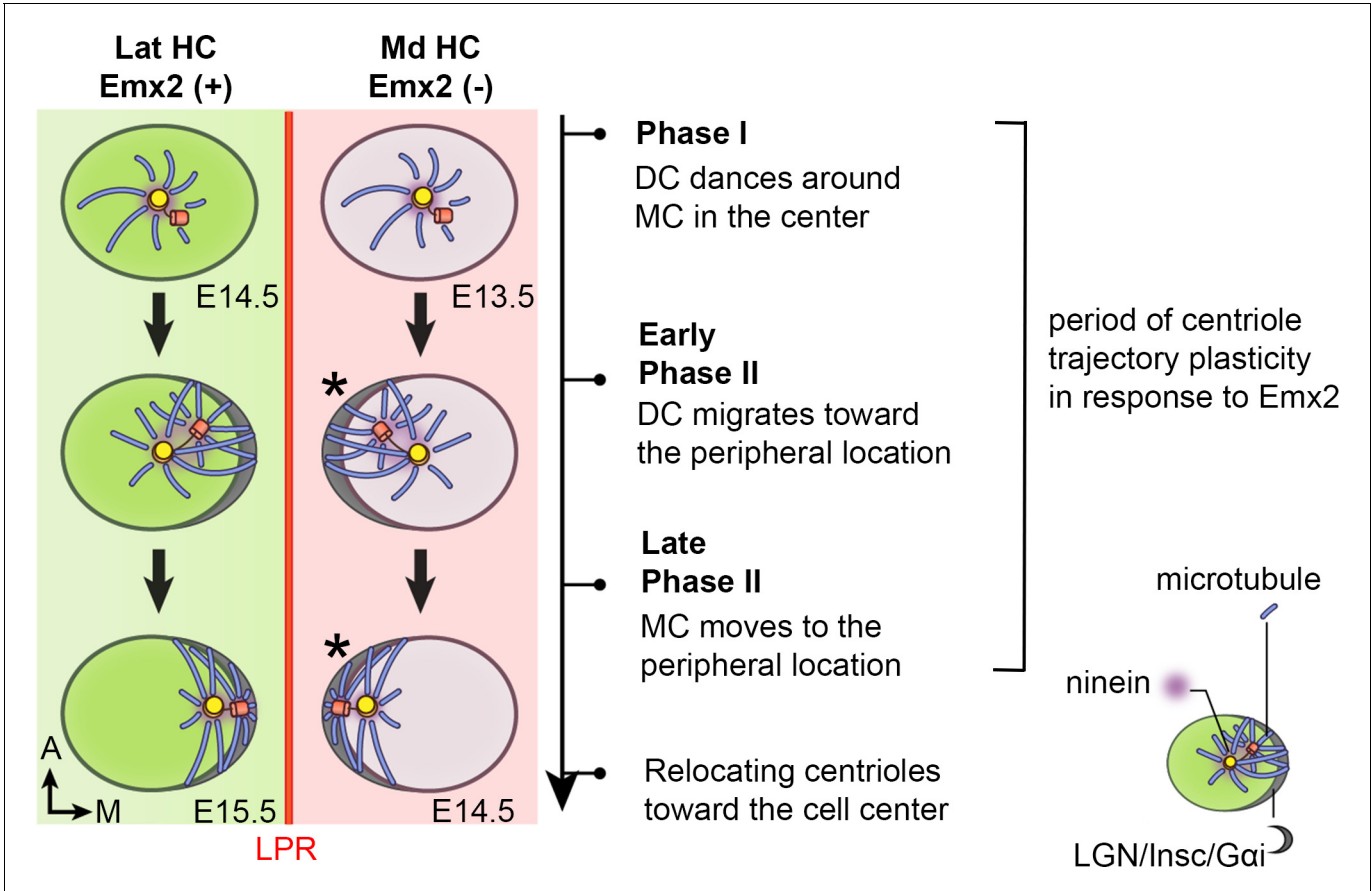

**Figure 10.** Summary of centriole trajectory during hair bundle establishment in nascent utricular HCs. In Emx2-negative Md HCs, Phase I is represented by MC located in the apical center of HCs with the DC dancing around the MC. Ninein is associated with the centrioles, which serves as the nucleation center for microtubules. In early Phase II, when DC migrates toward the peripheral side, the MC starts to follow the direction of the DC and the peripheral crescent LGN/Insc/Gαi complex starts to be established (*Ezan et al., 2013*; *Tarchini et al., 2013*). The broad distribution of ninein surrounding both centrioles is likely to anchor the minus end of microtubules and facilitates centriole migration to the periphery. By the end of Phase II, both MC and DC are located in the periphery and ninein becomes restricted to the centrioles again. Then, centrioles are relocalized towards the cell center by the bare zone (*Tarchini et al., 2013*). Cuticular plate starts to form at the beginning of Phase II based on β2-spectrin staining (not drawn). Centriole trajectory in both Phase I and early Phase II are reversible in the presence of Emx2 but responsiveness to Emx2 decreases over time (*Jiang et al., 2017*). Emx2-positive Lat HCs (green) show similar but opposite trajectory pattern of centriolar migration. Nevertheless, LGN/Insc/Gαi may be dispensable for hair bundle establishment in Md HCs since blocking Gαi with pertussis toxin does not appear to affect bundle orientation (asterisk).

ninein may function as microtubule anchors in addition to microtubule nucleation during this stage of centriole migration (*Figures 6* and *10*).

## Relationship between DC and MC migration

Many well described morphological and functional features distinguish between the MC and DC (*Pelletier and Yamashita, 2012*; *Fujita et al., 2016*). In addition to the function of DC maturing into an MC during the cell cycle, the function of the DC in differentiating cells is only beginning to be understood (*Loukil et al., 2017*; *Betleja et al., 2018*; *Gottardo et al., 2015*). Other than the DC being actively inhibited to form the cilium, recent results indicate that the proximity of the DC to the MC is also important for primary cilium formation (*Loukil et al., 2017*). Here, in nascent HCs, we show that when the MC was located at the center during Phase I, the DC was observed to move sporadically around the relatively stationary MC (*Figures 2* and *10*). This phenomenon has been described in several vertebrate somatic cell lines (*Piel et al., 2000*). In these cell lines, the MC, which is associated with a microtubule network is stationary, whereas the DC, although associated with the microtubule nucleation marker, γ-tubulin, is more mobile. The functional significance of the mobile

DC, however, remains unclear except that this behavior is regulated by the cell cycle and attenuates as cells transition from G1 to S phase.

During Phase II of the centriole migration, the DC invariably moved ahead of the MC to reach the peripheral destination. This pattern of the DC preceding the MC in migration was observed repeatedly in HCs under all conditions investigated such as during normal centriole migration, recovery from nocodazole treatments, and ectopic *Emx2* activation, suggesting that the migration of the DC is related to the MC. Although little is known about the relationships between the DC and MC in HCs, an inner ear conditional knockout of *Kif3a*, which encodes an intraflagellar transport protein, shows misplaced location and relationship between the MC and DC in cochlear HCs (*Sipe and Lu, 2011*).

Several scenarios could account for the observed behavior of the DC moving ahead of the MC in HCs. One possibility is that the two centrioles move independently of each other. Since the DC is similar to the MC in all parameters examined (association with the microtubule nucleation protein γ-tubulin and responses to microtubule disruption), each centriole can be independently transported by the microtubule-dynein system to their destinations in the peripheral cortex. The faster and higher mobility of the DC may simply be due to the MC being restricted by the attached cilium (*Paintrand et al., 1992*). An alternative scenario is that the MC is being dragged to the periphery by the DC via the intercentrosomal linkers between MC and DC, which are made of rootletin filaments (*Yang et al., 2006*). The sporadic movements of the DC around the MC in Phase I could also be regulated by the intercentrosomal linkers. In other systems, the length of these linkers can change and disintegrate based on maturation of the centrioles during the cell cycle (*Bahmanyar et al., 2008*; *Mardin et al., 2010*). However, little is known about the regulation and possible functions of these linkers in post-mitotic cells including HCs. Based on our findings, we speculate that the DC has an active role in guiding the MC/kinocilium to its proper location in differentiating HCs, in addition to its known role in regulating ciliogenesis.

## The role of Emx2 in reversing hair bundle orientation

In zebrafish lateral line, Emx2 regulates neuronal selectivity as well as hair bundle orientation (*Ji et al., 2018*). In the mouse utricle, onset of *Emx2* expression is well ahead of the emergence of HCs (*Figure 2—figure supplement 5*). Therefore, Emx2 may have a role in regional patterning and/or HC fate specification that indirectly lead to hair bundle reversal. This notion is supported by our live-imaging results demonstrating that HCs are already pre-patterned by Emx2 prior to centriole movements (*Figure 2*). Nevertheless, our ectopic *Emx2* experiments using AAV indicate that tdTomato signal was detectable within 36 hr of AAV-Emx2 infection (*Figure 5*). This time frame of sequential transcriptional and translational events to yield detectable tdTomato signal is comparable to other mammalian systems. Under the assumption that Emx2 is synthesized in a comparable time frame as tdTomato, hair bundle orientation reversal occurred relatively quickly within approximately 12 hr of detectable tdTomato (*Figure 5*). These results suggest that while Emx2 may have other functions in the utricle, its bundle reversal effect is likely to be direct and does not require multiple cascades of transcriptional and translational events. Additionally, both the genetic and AAV viral approaches indicate that during these early phases of centriole migration in hair bundle establishment, the system is plastic and responsive to Emx2 (*Figures 3* and *5*). However, the time-window of centrioles' responsiveness to *Emx2* is limited in vivo as ectopic activation of *Emx2* after E15.5 only has a moderate effect on hair bundle reversal in naive HCs (*Jiang et al., 2017*). Taken together these in vivo results and the in vitro results here showing a lack of correlation between the level of Emx2 expression and change in hair bundle orientation of AAV-Emx2-tdT infected HCs (*Figure 4*), suggest that there is a critical time-window when HCs are responsive to Emx2 (*Figure 10*). Our results suggest that ectopic Emx2 is able to alter the course of centriole trajectory during centriole migration (*Figures 3* and *5*, *Figure 3—figure supplement 2*) but perhaps as cuticular plate starts to form with the accumulation of β2-spectrin, centrioles are hindered from relocation (*Figure 9*).

Furthermore, our results showed that the DC may have an active role in guiding the MC to its designated location in the HC periphery and a positive force is required to actively maintain this peripheral centriole positioning. These findings provided insights into the regulation of centriole dynamics during hair bundle establishment.

# Materials and methods

## Key resources table

| Reagent type (species) or resource | Designation | Source or reference | Identifiers | Additional information |
|---|---|---|---|---|
| Genetic reagent (*M. musculus*) | CAG:GFP-Centrin2 | PMID:21752934 | RRID:MGI:3793421 | Xiaowei Lu (University of Virginia) |
| Genetic reagent (*M. musculus*) | Atoh1-Cre | PMID:19609565 | RRID:MGI:3775845 | Bernd Fritzsch (University of Iowa) |
| Genetic reagent (*M. musculus*) | Gfi1-Cre | PMID:20533399 | RRID:MGI:4430258 | Lin Gan (Augusta University) |
| Genetic reagent (*M. musculus*) | Rosa26R$^{Emx2}$ | PMID:28266911 | | |
| Genetic reagent (AAV-Virus) | AAV2.7m8-CAG-Emx2-P2A-tdTomato | Vector Biolabs | | $4.0 \times 10^{10}$ GC in 100 µl culture medium |
| Genetic reagent (AAV-Virus) | AAV2.7m8-CAG-tdTomato | Vector Biolabs | | $4.0 \times 10^{10}$ GC in 100 µl culture medium |
| Chemical compound, drug | nocodazole | Sigma-Aldrich | M1404 | 5 µM |
| Chemical compound, drug | pertussis toxin | Millipore Sigma | 516560–50 UG | 8.4 nM |
| Chemical compound, drug | SiR-tubulin | Spirochrome | SC002 | 1 µM |
| Antibody | Mouse anti-βII spectrin | BD Biosciences | Cat# 612562, RRID:AB_399853 | IHC (1:500) |
| Antibody | Rabbit anti-Arl13b | Proteintech | Cat# 17711–1-AP, RRID:AB_2060867 | IHC (1:500) |
| Antibody | Mouse anti-acetylated tubulin | Sigma-Aldrich | Cat# T7451, RRID:AB_609894 | IHC (1:500) |
| Antibody | Rabbit anti-ninein | Abcam | Cat# Ab231181 | IHC (1:500) |
| Antibody | Rabbit anti-γ-tubulin | Sigma Aldrich | Cat# ab4447, RRID:AB_304460 | IHC (1:500) |
| Antibody | Rabbit anti-Emx2 | Trans Genic | Cat# KO609 | IHC (1:250) |
| Antibody | Rabbit anti-Gαi | Provided by B. Nurnberg | | IHC (1:1000) |
| Antibody | Alexa Fluor 488 donkey anti-mouse IgG | Thermo Fisher Scientific | Cat# A-21202 RRID:AB_2535788 | IHC (1:500) |
| Antibody | Alexa Fluor 647 donkey anti-mouse IgG | Thermo Fisher Scientific | Cat# A-31571 RRID:AB_162542 | IHC (1:500) |
| Antibody | Alexa Fluor 405 Donkey anti rabbit IgG | Abcam | ab175651 | IHC (1:500) |
| Antibody | Alexa Fluor donkey anti-rabbit 647 | Thermo Fisher Scientific | Cat# A-31573 RRID:AB_2536183 | IHC (1:500) |
| Software, algorithm | ImageJ | | RRID:SCR_003070 | |
| Software, algorithm | Imaris 9.5.0 | Bitplane | RRID:SCR_007370 | |

## Mouse

All animal experiments were conducted according to NIH guidelines and under the Animal Care Protocol of NIDCD/NIH (#1212–17). *CAG:GFP-Centrin2* mice were obtained from Xiaowei Lu at University of Virginia (RRID:MGI:3793421), *Atoh1-Cre* mice from Bernd Fritzsch at University of Iowa (RRID:MGI:3775845), and *Gfi1-Cre* mice from Lin Gan at Augusta University (RRID:MGI:4430258; *Yang et al., 2010*). The *Rosa26R$^{Emx2}$* mouse was generated by knocking in the cassette *attb-pCA promoter-lox-stop-lox-Emx2-T2A-Gfp-WPRE-polyA-attb* to the *Rosa* locus as described previously

(*Jiang et al., 2017*). *Rosa26R$^{tdTomato}$* were purchased from Jackson laboratory (RRID:IMSR_JAX:007914, *Madisen et al., 2010*). *Atoh1$^{Cre}$; Rosa$^{tdT/+}$* control specimens for live imaging was generated by crossing *Atoh1$^{Cre}$; Rosa$^{tdT/tdT}$* males with *CAG:GFP-Centrin2$^{+/-}$* females. Emx2 gain-of-function specimens was generated by crossing *Atoh1$^{Cre}$; Rosa$^{tdT/tdT}$* or *Gfi1$^{Cre}$; Rosa$^{tdT/tdT}$* males with *CAG:GFP-Centrin2$^{+/-}$; Rosa$^{Emx2/Emx2}$* females. In addition to tdTomato signals, HC identity in the live-images was confirmed based on the round or oval shape of the apical cell surface and its more apical position of the nucleus within the epithelium relative to nuclei of the supporting cells.

## Live imaging

The mouse utricle together with anterior and lateral cristae for orientation were dissected from E13.5 or E14.5 mouse inner ears. The harvested tissue was mounted on a Cell-Tak (Corning, NY, NY)-coated coverslip (*Belyantseva, 2016*) in culture medium containing DMEM/F12 (Thermo Fisher Scientific, Waltham, MA), 10% of fetal bovine serum (FBS, Thermo Fisher Scientific, Waltham, MA) and 50 U/ml penicillin G (Sigma-Aldrich, St Louis, MO), unless indicated otherwise. Live imaging was started after the explant had been incubated for 4 hr in a tissue culture incubator to ensure full attachment of the explant to the coverslip. The imaging was conducted in a chamber maintained at 37°C and 5% $CO_2$ on either an inverted PerkinElmer UltraVIEW Time Lapse Image Analysis System with a CMOS camera or a Nikon A1R HD confocal system on a Ni-E upright microscope with a GaAsP detector. For the UltraVIEW, a 10x objective was used for the lower magnification images and a 63x objective was used for time-lapse imaging (pixel size is 0.216 × 0.216 μm). For the Nikon A1R, a 25x objective was used for both the low- (pixel sizes are 0.48 × 0.48 μm) and high-magnification (pixel sizes 0.16 × 0.16 μm) time lapse recordings. In both microscopes, Z-stacks of 30–60 μm thickness with a 0.5 μm step were taken at each time frame per 10 min intervals. Live imaging was conducted up to 41 hr.

For microtubule inhibition experiments, we first determined the dose of nocodazole (Catalog# M1404, Sigma-Aldrich, St Louis, MO) to apply. Two doses of nocodazole, 5 μM and 33 μM, which were used in other cochlear explant studies (*Szarama et al., 2012*; *Shi et al., 2005*), were tested on E13.5 utricular explants for 24 hr. While explants treated with 5 μM nocodazole showed largely intact HCs with reduced tubulin signals in the cytoplasm, the utricular explants treated with 33 μM nocodazole showed reduced number of HCs, which looked unhealthy or apoptotic. Therefore, 5 μM of nocodazole was chosen for our live imaging studies. Nocodazole in DMEM/F12 with 10% FBS or culture media used for washing out the nocodazole was added to the utricular explant directly without disturbing the position of the explant under the microscope.

For the pertussis toxin experiments, we tested two dosages of pertussis toxin (Catalog# 516560–50 UG, Millipore Sigma, St Louis, MO), 2.1 and 8.4 nM, that were used for an in vivo inner ear study (*Kakigi et al., 2019*) and an in vitro outer HC study (*Kakehata et al., 1993*), respectively. *Figure 9—figure supplement 1a* shows the results of 8.4 nM pertussis toxin treatments, but the phenotypes observed were consistent between both dosages.

## AAV virus

The AAV2.7m8-CAG-Emx2-P2A-tdTomato (2.2 × 10$^{12}$ genome copies/ml (GC/ml)) and AAV2.7m8-CAG-tdTomato (5.4 × 10$^{12}$ GC/ml) were synthesized by Vector Biolabs, Inc. The expression of *Emx2* and *tdTomato* were driven by the CAG promoter. Four hr after an E13.5 utricular explant was established on a Cell-Tak coated coverslip in culture medium, the coverslip was transferred to a 24-well tissue culture plate filled with 100 μl of DMEM/F12 containing 2% of FBS and AAV in a final concentration of 4.0 × 10$^{10}$ GC. One hour later, another 100 μl of medium without AAV was added to the culture. After an overnight incubation, the culture was washed several times before changing to regular medium that contained 10% FBS.

## Whole mount immunostaining

Dissected or live-imaged utricles attached on Cell-Tak coated coverslips were fixed with 4% paraformaldehyde in PBS at room temperature for 15 min. After fixation, the tissue attached to the coverslip was washed three times in PBS before blocking with PBS containing 5% donkey serum and 0.3% Triton-X for 45 min. For primary antibody, mouse anti-βII spectrin (1:500; BD Biosciences, San Jose, CA, Catalog# 612562, RRID:AB_399853), rabbit anti-Arl13b (1:500; Proteintech, Rosemont, IL,

Catalog# 17711–1-AP, RRID:AB_2060867), mouse anti-acetylated tubulin (1:500; Sigma-Aldrich, St Louis, MO, Catalog# T7451, RRID:AB_609894), rabbit anti-ninein, (1:500; Abcam, Cambridge, UK, Catalog# Ab231181), rabbit anti-γ-tubulin (1:500; Sigma-Aldrich, St Louis, MO, Catalog# ab4447, RRID:AB_304460), rabbit anti-Emx2 (1:250; KO609, Trans Genic, Fukuoka, Japan), and rabbit anti-Gαi (1:1000; provided by B. Nurnberg) was used for overnight incubation at 4°C. Secondary antibodies of either Alexa Fluor 488/647 donkey anti-mouse IgG (Thermo Fisher Scientific, Waltham, MA, Catalog# A-21202 RRID:AB_2535788/Catalog# A-31571 RRID:AB_162542), Alexa Fluor 405 Donkey anti rabbit IgG (ab175651, Abcam, Cambridge, MA), or Alexa Fluor donkey anti-rabbit 647 (Thermo Fisher Scientific, Waltham, MA, Catalog# A-31573 RRID:AB_2536183) was used at 1:500 dilution for 15 min at room temperature. After extensive washing with PBS, samples were mounted in ProLong Gold Antifade (Thermo Fisher Scientific, Waltham, MA).

For experiments correlating live-imaged HCs followed by anti-β2-spectrin and anti-Gαi immunostaining, explants were fixed immediately after 2 hr of live imaging and processed for immunostaining. Then, the final frame of live imaging at low power was compared to the confocal image of the immunostained specimen. Based on the comparable distribution pattern of tdTomato-positive HCs between the two images, individual HCs tracked in the live imaging recording was identified and correlated with their immunostaining in the confocal images.

SiR-tubulin (Catalog# SC002, Spirochrome, Switzerland) was added to the live utricular samples attached on the coverslip to achieve a final concentration of 1 μM in the culture medium containing 10% FBS in DMEM/F12 and incubated for 10 hr before imaging. For imaging of γ-tubulin, acetylated-tubulin, and ninein staining as well as SiR-tubulin labeling, Airyscan imaging was conducted on a Zeiss LSM 780 with an Airyscan attachment (Carl Zeiss AG, Oberkochen, Germany) using a 63 × 1.4 NA oil objective lens. The acquired images were processed by Zeiss Zen Black software v2.1 for deconvolution.

## In situ hybridization

In situ hybridization was conducted as described previously (*Morsli et al., 1998*). Digoxigenin-labeled RNA probes were synthesized for *Sox2* (*Evsen et al., 2013*) and *Emx2* (*Simeone et al., 1992*) as described.

## Analysis

To correct for sample drifting during imaging over time, the 3-D time-lapse image was processed using ImageJ (RRID:SCR_003070 *Schneider et al., 2012*) and the plugin of PoorMan3DReg (http://sybil.ece.ucsb.edu/pages/software.html) was used to re-align HCs to a stabilized x-y positions after semi-automatic adjustment of the z-positions using ImageJ macro, manual z-stack regulator (provided by Dr. Sho Ota). In some utricular images, GFP signals that overlapped with tdTomato signals in HCs were segmented and GFP signals without overlapping tdTomato signals were eliminated by using the ImageJ macro, Centrin-detector (provided by Dr. Sho Ota). The processed images were used for subsequent analyses. For tracking of centrioles and determination of the apical center of HCs, we performed spot and cell tracking algorithm using Imaris 9.5.0 (RRID:SCR_007370, Bitplane, Zurich, Switzerland) and these tracking data were used to generate the 3D reconstructed videos.

Three-dimensional reconstruction of the HC shape, total centriole trajectory relative to the center of the HC as well as temporal trajectory of selected time-frames of the recordings were generated by tracking data using Imaris. HC shape was determined based on tdTomato expression. For most summaries of selected time frames where the ending positions of the centrioles were drawn as circles such as *Figure 2D*#2, the HC contour of the last time frame was used. In selected time frames where the initial positions of the centrioles were shown as triangles such as *Figure 2D*#1, the HC contour at the beginning of the recording was used. In cases where the cell center was undefinable due to absence of tdTomato signal as in progenitors and supporting cells, or in young Lat HCs and AAV-infected Md HCs before clear detectable tdTomato expression, DC positions were plotted relative to the MC, which was used as a proxy for the center of the cell and thus no cell shape was drawn (*Figure 2G*#1, *Figure 2—figure supplement 3B*#1, *Figure 2—figure supplement 4C–E*, *Figure 5C–D*).

The entry of centriole movements into Phase II was determined retroactively when the DC is consistently moving toward the peripheral direction where hair bundles should be established. The x-y and z distance between centrioles were measured based on the coordinate of the tracking data. For the calculation of average x-y and z distances of centrioles in Lat HCs, only last 30 time-points before entering Phase II was used for computing the averages of Phase I Lat HCs, and earlier time-points possibly as progenitors were eliminated. The x-y migration speed was measured from the difference of coordinates of each centriole compared to the center of the HC per each 10 min time-frame, and the average speed of each centriole was calculated from all the time points after HC center could be determined based on tdTomato signals.

The tdTomato expression level of AAV-Emx2-tdTomato infected HCs was measured using ImageJ based on the relative fluorescent intensities of the tdTomato in HCs subtracted by signals from the surrounding background. To measure ninein fluorescence intensity associated with the centrioles, a line was drawn connecting the two centrioles and ninein signals associated with the centrioles (based on centrin-gfp signals) and in between the centrioles were measured along this line using ImageJ. For measuring the distribution of ninein during centriole migration, the ninein-positive region above threshold was outline and measured using Image J, and differences in ninein areas between Phases were analyzed by one-way ANOVA. Direction of DC trajectories in HCs ectopically expressing *Emx2* using either *Atoh1cre* or *Gficre* was analyzed by Chi-square test and for other statistics, students t-test was used. P-values of less than 0.05, 0.01 and 0.001 are indicated by \*, \*\* and \*\*\*, respectively.

## Acknowledgements

We thank Dr. Sho Ota for writing the Fiji macro and providing helpful advice on live imaging data analyses, Kevin Isgrig and Dr. Wade Chen at NIDCD for advice on AAV2.7m8 virus and Dr. Elizabeth Driver at NIDCD for help with live imaging. We are also grateful to Drs. Brian Galletta, Matt Hannaford and Nasser Rusan at National Heart Lung and Blood Institute and Drs. Elizabeth Driver, Inna Belyantseva and Ronald Petralia at NIDCD, and members of the Wu Lab for their critical review of the manuscript. We also want to thank Michael Mulheisen for technical support in conducting the in situ hybridization experiments and Erina He at the NIH Medical Arts for drawing the illustrations.

## Additional information

### Competing interests
Doris K Wu: Reviewing editor, *eLife*. The other author declares that no competing interests exist.

### Funding

| Funder | Grant reference number | Author |
|---|---|---|
| National Institutes of Health | 1ZIADC000021 | Doris K Wu |
| Japan Society for the Promotion of Science | JSPS Research Fellowship for Japanese Biomedical and Behavioral Researchers at NIH | Yosuke Tona |

The funders had no role in study design, data collection and interpretation, or the decision to submit the work for publication.

### Author contributions
Yosuke Tona, Conceptualization, Data curation, Software, Formal analysis, Validation, Investigation, Visualization, Writing - original draft, Writing - review and editing; Doris K Wu, Conceptualization, Supervision, Funding acquisition, Methodology, Writing - review and editing

## Author ORCIDs

Yosuke Tona (iD) https://orcid.org/0000-0001-6868-8057
Doris K Wu (iD) https://orcid.org/0000-0002-1400-3558

## Ethics

Animal experimentation: All animal experiments were conducted according to NIH guidelines and under the approved Animal Care Protocol of NIDCD/NIH (#1212-17).

## Decision letter and Author response

Decision letter https://doi.org/10.7554/eLife.59282.sa1
Author response https://doi.org/10.7554/eLife.59282.sa2

## Additional files

### Supplementary files

• Transparent reporting form

### Data availability

The following figures contain the source data files. Figure 2 (source data 1), Figure 2 supplement 2 (source data 1-3), Figure 2 supplement 3 (source data 1-4), Figure 2 supplement 4 (source data 1-2), Figure 3 (source data 1-2), figure 3 supplement 2 (source data 1), Figure 4 (source data 1-2), Figure 5 (source data 1), Figure 6 (source data 1-2), Figure 7 (source data 1), Figure 7 supplement 1 (source data 1), Figure 8 (source data 1-2), Figure 9 (source data 1).

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
