## [Decision Letter]

Thank you for submitting your article "Live imaging of hair bundle polarity acquisition demonstrates a critical timeline for transcription factor Emx2" for consideration by *eLife*. Your article has been reviewed by three peer reviewers, and the evaluation has been overseen by a Reviewing Editor and Marianne Bronner as the Senior Editor. The following individuals involved in review of your submission have agreed to reveal their identity: Mireille Montcouquiol (Reviewer #2); Raj K Ladher (Reviewer #3).

The reviewers have discussed the reviews with one another and the Reviewing Editor has drafted this decision to help you prepare a revised submission.

Summary:

The study is the first live-imaging analysis of centriole dynamics in developing mouse utricular hair cells before and during hair bundle formation. The centrifugal migration of the basal body towards the hair cell periphery is thought to be the symmetry breaking event that determines hair bundle polarity and orientation. Two major findings are reported. First, the daughter centriole exhibits higher motility and migrates earlier than the basal body and may play an active role in kinocilium migration. Second, the transcription factor Emx2, which is necessary and sufficient to instruct lateral hair cell (LHC) polarity, can repolarize medial hair cells (MHCs) during the window of active centriole motility. The conclusions were supported by rigorous quantitative analysis, as well as pharmacological and genetic manipulations.

The reviewers all agreed that the manuscript represents an impressive technical advance in the live imaging of centriole migration during sensory hair cell development, and fits very nicely into the 'Research Advance' category. However, the reviewers would also like some further analysis of the mechanisms involved. The reviewers and editors appreciate that these are technically challenging experiments, but hope that many of the comments can be addressed by further analysis, quantification or re-plotting of existing data, together with some additional discussion. A few points may require additional experimental work.

Essential revisions:

1) Provide further analysis of the processes that influence the plasticity period of PCP establishment (centriole motility post-Phase II, and the relationship between cuticular plate development and kinocilium migration (reviewer 1, points 1, 7, 12).

2) Strengthen support for the conclusion that daughter centriole movement precedes migration of the mother centriole during polarisation and repolarisation (reviewer 1 comments, 4, 5).

3) The data on Ninein localisation (Figure 6) should be strengthened, e.g. by showing a condition that altered ninein localisation during centriole migration (reviewer 1, point 8; reviewer 2, paragraph 5).

4) Provide additional discussion on the levels of Emx2 (reviewer 2, paragraph 6).

5) Address the queries relating to microtubule behaviour and dynamics, and the effects of nocodazole on hair bundle polarity (reviewer 1, points 9-11).

6) Consider and discuss the potential role of actin in centriole migration.

7) Address the request from reviewer 3 asking about supporting cell centriole behaviour, in order to distinguish any hair-cell-specific centriole behaviours from those that are generic. If data are available to address this point, please include these – if not, comment and discuss.

The full reviews are appended below for your information.

Reviewer #1:

While this study describes intriguing hair cell centriole behaviors and further confirms Emx2's role in pre-patterning of LHC polarity, mechanistic insights into regulation of centriole motility/migration or how Emx2 instructs LHC polarity would require further experiments.

1) Since live imaging can be extended up to 41 hours, it would be informative to examine MHC centriole behaviors post-Phase II. Do centrioles become stationary once they reach the hair cell periphery? Or do they remain motile? How long is the "plastic" phase of centriole motility? This may be addressed by time-lapse more mature MHCs in E14.5 explants.

2) It was unclear whether the Z distances were included in calculating the inter-centriolar distances. Even though the Z resolution of the microscope was limited, it would still be informative to determine whether the motility of mother (MC) and daughter hair cell centrioles (DC) is largely planar (in the x-y dimension) or three dimensional. It seems plausible that the x-y distance changes while the 3D distance is maintained.

3) Figure 2—figure supplements 1 and 2. Quantifications of two MHCs and two LHCs showed that LHCs appeared to have a longer Phase I (20 hours) compared to MHCs (4-8 hours). Is this difference significant? Or could it be explained by the more mature state of MHCs at time "0" compared to LHCs? Or that LHCs were unhealthy at time "0" due to prolonged imaging? Would imaging LHCs in E14.5 explants be more comparable to MHCs at E13.5?

4) Subsection “Migration of DC precedes MC in medial utricular HCs during hair bundle establishment”. It was stated that DC directional migration precedes that of MC in Phase II. It would be important to show supporting data (e.g. plotting temporal changes in MC and DC positions in addition to showing their cumulative trajectories).

5) Subsection “Reversing hair bundle orientation by ectopic Emx2”. Likewise, it would be important to show data demonstrating that counter-migration of DC precedes reorientation of MC during MHC repolarization, by tracking DC and MC movement during repolarization over time. It would lend further support to the idea that DC actively participate in kinocilium migration.

6) Since it is not known how long Phase II lasts, it is uncertain that the peripheral location of the centrioles indicated "that these cells were at the end of Phase II".

7) In fact, the AAV-Emx2 transduced MHCs provide an excellent opportunity to compare and contrast centriole behaviors between repolarized and non-repolarized MHCs. This would determine whether centriole behaviors around the time of Tomato expression could predict repolarization competence, and shed light on whether centriole motility underlies the repolarization "plasticity" window.

8) Figure 6C. Ninein staining around the centrioles at different phases of centriole migration should be quantified to demonstrate the "broad" vs. "focused" distribution pattern.

9) Figure 7 shows one example of destabilized centriole position upon Nocodazole treatment and centriole re-positioning following washout. Since treated MHCs were at different stages of polarization, it should be both feasible and informative to assess a number of treated cells and determine whether and how many (a) Phase I cells proceeded to Phase II, (b) Phase II cells reverted to Phase I (as shown in Figure 7C), and (c) post-Phase II cells with unaffected centriole position in the presence of Nocodazole. This would more convincingly demonstrate the role of microtubules in kinocilium migration.

10) Given that the role of kinocilium in utricular hair bundle polarity is debated, it would be important to interfere with microtubules and assess hair bundle polarity. Does prolonged treatment of utricle explants with either nocodazole or SiR-tubulin (which can stabilize microtubules) affect hair bundle polarity?

11) Related to above, studies of auditory hair cells by Ezan et al. suggest that microtubule cortical capture is critical for kinocilium migration. Thus it would be important to test whether blocking microtubule dynamic instability (without destroying microtubules per se) using low-dose nocodazole would disrupt Ninein localization, Phase I and Phase II centriole behaviors and utricle hair bundle polarity.

12) The relationship between cuticular plate formation and kinocilium migration was not addressed and warrants further analysis. Correlating anti-spectrin staining with centriole position in hair cells at early stages of polarization (e.g. Phase I, Phase II and post-Phase II equivalent) would inform how the developing cuticular plate might physically constrain centriole migration and impede repolarization competence.

Reviewer #2:

In this manuscript, the authors record and analyze the dynamic behavior of the daughter and mother centrioles associated with the future single kinocilium at the surface of the hair cells of the vestibular epithelia, just before and at the onset of PCP establishment. The authors confirm a previous claim of a directed movement of the kinocilium/centrioles via microtubules and demonstrate that whether or not the hair cell is present in an Emx2-positive region of the epithelium, the overall dynamic of the centrioles is similar. Finally, they also show that there is a window of opportunity to re-route the centriole in an Emx2-dependant manner.

This is a nice and interesting study that expands our knowledge of how PCP is established early in these cells, and notably brings a dynamic and timing element to the mechanism. The results are convincing, and the conclusions not overstated, and I believe it would be of interest not only for scientists in the inner ear field but also for scientists in the PCP and the cilium communities. I must emphasize the technical difficulty of such a study, not only because of the number and complexity of the transgenic mice used but also because time-lapses experiments are not so common in this system due to its size and difficulty of access.

The authors mention an interesting study by the group of Michel Bornens (Piel et al., 2000) where Piel and colleagues evaluate the role of the cytoskeleton on centrioles movement. These authors notably show that the "characteristic motions of the daughter centriole persist in the absence of microtubules (Mts), or actin, but are arrested when both Mts and actin filaments are disrupted." Tona and Wu use transient nocodazole treatment to disrupt MT and as a result observed a return of the centrioles to the center of the hair cell and an increase in DC movement. But is actin also participating in the peripheral movement of the centrioles and/or its maintenance at the periphery? The authors seem to suggest that it is only MT dependant.

In that same paper, the authors measured the area covered by the centriole as a measure of the persistent motion exhibited by a centriole (Figure 5C of that paper). Would it be possible, with the data already at hand, to extract an area covered by the centriole, with respect to HC surface? Or at the very least to highlight a bit better the typical trajectory of a MC and a DC? The trajectories as illustrated in Figure 1D in yellow and orange are difficult to follow. Though the corresponding videos are quite clear, it would be nice to have a schematic with an expended HC surface and different colors to better follow.

I am not entirely convinced by the conclusions from Figure 6. It is a really informative piece of data but despite a number of cells recorded displaying what seems to be a similar profile of expression of ninein, the staining is spread out and uneven. I would point to Figure 6C Phase II (early), the ninein staining, though covering both centrioles, seems stronger/enriched at the level of the MC. Is it always true? Same observation for image Phase II (late). Is there a way to illustrate this with a heat map or some sort of quantification?

The authors speculate on temporal sensitivity of the hair cells in response to Emx2 in different contexts: the transgenic knockout mice, the overexpression mice, and the AAV. They discuss this in the subsection “The role of Emx2 in reversing hair bundle orientation”, always in terms of timing. I am wondering if the levels of Emx2 could not participate in the overall mechanism. It is well known for example that core PCP signaling needs to be finely controlled, as too much or too little of core PCP protein will affect the polarity of the tissue. Could the authors speculate, based on what is known about Emx2 in other species and/or tissues, regarding this aspect? I agree that the 3 days delay between the detection of Emx2 in the future sensory epithelium and the detection of hair cells could reflect other roles of Emx2, but could it also be necessary for some accumulation? From a quick survey of the literature, it seems a concentration-dependent mechanism has been suggested for Emx2 in cortical progenitors.

Regarding the discussion on "pulling" versus "shortening" of the MT (subsection “Hair bundle establishment in HCs”), I do not think the two are mutually exclusive and that one should prevail. Even if the authors have supporting data for the former, the latter is probably also at play. The simplest reason being that the accumulation of the MT fibers has to somehow be "evacuated" as it is pulled, to prevent its accumulation on one side of the HC.

Reviewer #3:

The cellular mechanisms that drive the deflection of the kinocilia during generation of polarity are unknown. In the submission from Tona and Wu, they use some lovely live cell imaging to understand the migration of the mother and daughter centriole in the establishment of the polarised hair cell.

They find that the daughter centriole precedes the mother, in the migration to one side of the cell. Emx2, which seems to reverse the polarity of utricular hair cells, seems to also reverse the migration of the daughter centriole, suggesting there are intrinsic propensities within the hair cell that enable the centriolar response to polarity generation signals.

I like the story very much, and I think that there is the potential to have very profound insights into the process of polarity generation in the inner ear. As it is, there are more questions raised that I would like to see answered.

Immunostaining of the centrioles should be performed. In particular looking a centrin (mother and daughter), Odf2 (mother-specific), PARP3 and centrobin (daughter-enriched) should be done. Ideally this would be done with Gai staining as well, to ask if the position of the daughter centriole is coordinated.

Supporting cells also have centrioles that will generate the basal body for their cilia. What are the supporting cell centrioles doing? Are they coordinated as well? Does Emx2 change their migration?

One of the obvious parallels is that with spindle orientation. It is surprising that the Gai/LGN system was not directly investigated, although the insinuation is that this data is available (in Figure 8?). What is migration like when the explant is treated with pertussis toxin.

---

## [Author Response]

Essential revisions:1) Provide further analysis of the processes that influence the plasticity period of PCP establishment (centriole motility post-Phase II, and the relationship between cuticular plate development and kinocilium migration (reviewer 1, points 1, 7, 12).

As suggested by the reviewers, we correlated the centriole trajectory that we described with cuticular plate formation using anti-spectrin antibody as well as the polarity complex, LGN/Ins/Gαi, required for hair bundle establishment using anti-Gαi antibody. These results are now shown in Figure 9 of the revised manuscript. Our results show that Gαi expression is diffused initially and as the centrioles start to migrate to the periphery (beginning of Phase II), Gαi and spectrin staining start to show restricted expression and these complementary expression patterns become well defined by late Phase II.

2) Strengthen support for the conclusion that daughter centriole movement precedes migration of the mother centriole during polarisation and repolarisation (reviewer 1 comments, 4, 5).

We have included additional temporal trajectory relationship between the two centrioles from the end of Phase I to early Phase II showing that the daughter centriole migrated ahead of the mother centriole during normal centriole migration and in response to ectopic Emx2 as well as recovery from nocodazole treatments. These results are shown in Figures 2, 3, 5 and 7 and Figure 1—figure supplement 3 and Figure 7—figure supplement 1.

3) The data on Ninein localisation (Figure 6) should be strengthened, e.g. by showing a condition that altered ninein localisation during centriole migration (reviewer 1, point 8; reviewer 2, paragraph 5).

Per the reviewers’ request, we have quantified the distribution of ninein expression during Phase I and Phase II. The quantification results support a difference in ninein distribution during centriole trajectory to the periphery. The association of ninein with specific centrioles during Phase I or Phase II, however, is not consistent among hair cells. See results of Figure 6 for details.

4) Provide additional discussion on the levels of Emx2 (reviewer 2, paragraph 6).

Discussion of Emx2 has been added to the Results and Discussion (see Figure 4).

5) Address the queries relating to microtubule behaviour and dynamics, and the effects of nocodazole on hair bundle polarity (reviewer 1, points 9-11).

We have conducted additional experiments and quantified our existing results showing that nocodazole affects centrioles during centriole migration but under our experimental conditions, it did not seem to affect centriole positioning by E14.5 when hair cells were more mature. See results in the new Figures 7 and 8.

6) Consider and discuss the potential role of actin in centriole migration.

We have done some experiments to address the role of actin in centriole migration. See our detailed explanation below.

7) Address the request from reviewer 3 asking about supporting cell centriole behaviour, in order to distinguish any hair-cell-specific centriole behaviours from those that are generic. If data are available to address this point, please include these – if not, comment and discuss.

We have analyzed the centrioles in the supporting cells and they basically stayed in the center without peripheral trajectory, unlike the hair cells. These results are now illustrated as a new Figure 2—figure supplement 4.

The full reviews are appended below for your information.Reviewer #1:While this study describes intriguing hair cell centriole behaviors and further confirms Emx2's role in pre-patterning of LHC polarity, mechanistic insights into regulation of centriole motility/migration or how Emx2 instructs LHC polarity would require further experiments.1) Since live imaging can be extended up to 41 hours, it would be informative to examine MHC centriole behaviors post-Phase II. Do centrioles become stationary once they reach the hair cell periphery? Or do they remain motile? How long is the "plastic" phase of centriole motility? This may be addressed by time-lapse more mature MHCs in E14.5 explants.

Per the reviewer’s suggestion, we analyzed our recordings beyond 20 hours for the Md HCs and we found that the two centrioles moved back towards the center of the HCs. Unlike other stages, the two centrioles appeared to move in unison towards the center of the hair cell. These results are consistent with the report by Tarchini et al., 2013, describing the formation of the bare zone which relocate the kinocilium toward the center of the HC to their final position prior to hair bundle establishment.

2) It was unclear whether the Z distances were included in calculating the inter-centriolar distances. Even though the Z resolution of the microscope was limited, it would still be informative to determine whether the motility of mother (MC) and daughter hair cell centrioles (DC) is largely planar (in the x-y dimension) or three dimensional. It seems plausible that the x-y distance changes while the 3D distance is maintained.

Our report of x-y distances did not include the Z distance. The quantification of Z distances has now been added to Figure 2—figure supplements 2 and 3. Interestingly, most Z-distance migrations occurred during Phase I and not Phase II when centrioles showed the biggest difference in the x-y distance. This observation applies to both medial and lateral hair cells.

3) Figure 2—figure supplements 1 and 2. Quantifications of two MHCs and two LHCs showed that LHCs appeared to have a longer Phase I (20 hours) compared to MHCs (4-8 hours). Is this difference significant? Or could it be explained by the more mature state of MHCs at time "0" compared to LHCs? Or that LHCs were unhealthy at time "0" due to prolonged imaging? Would imaging LHCs in E14.5 explants be more comparable to MHCs at E13.5?

Yes, the duration of Phase I of Lat HCs seemed longer than the Md HCs but it is due to a difference in the developmental timing between the two types of HCs rather than a difference in the duration of Phase I. In the medial utricle, some hair cells are born (post-mitotic) as early as E11.5, whereas the peak of cell cycle exit for hair cell precursors in the lateral utricle is not until E14.5 (Jiang et al., 2017). Therefore, most hair cells in the lateral utricle were not formed yet at the beginning of the live imaging and thus they seemed to spend a much longer time in Phase I than Md HCs. The delay in the appearance of tdTomato signal in the Lat HCs is consistent with this hypothesis.

4) Subsection “Migration of DC precedes MC in medial utricular HCs during hair bundle establishment”. It was stated that DC directional migration precedes that of MC in Phase II. It would be important to show supporting data (e.g. plotting temporal changes in MC and DC positions in addition to showing their cumulative trajectories).

In the revised manuscript, we have added additional color-coded, temporal trajectory during the end of Phase I and beginning hours of Phase II when DC migrated ahead of the MC in Md HCs (Figures 2, 3), Lat HCs (Figure 2, Figure 2—figure supplement 3) as well as conditions in which centriole trajectory was redirected such as hair cells overexpressing Emx2 (Figure 3), hair cells infected with Emx2-AAV (Figure 5), and hair cells recovering from nocodazole treatments (Figure 7).

5) Subsection “Reversing hair bundle orientation by ectopic Emx2”. Likewise, it would be important to show data demonstrating that counter-migration of DC precedes reorientation of MC during MHC repolarization, by tracking DC and MC movement during repolarization over time. It would lend further support to the idea that DC actively participate in kinocilium migration.

See response to #4 above. These results have been incorporated into the Results section.

6) Since it is not known how long Phase II lasts, it is uncertain that the peripheral location of the centrioles indicated "that these cells were at the end of Phase II".

We thank the reviewer for pointing out this oversight. This phrase has been fixed in the revised manuscript.

7) In fact, the AAV-Emx2 transduced MHCs provide an excellent opportunity to compare and contrast centriole behaviors between repolarized and non-repolarized MHCs. This would determine whether centriole behaviors around the time of Tomato expression could predict repolarization competence, and shed light on whether centriole motility underlies the repolarization "plasticity" window.

Unfortunately, bundle repolarization is not correlated with the intensity of tdTomato or Emx2 expression. This is consistent with our hypothesis that the stage of the hair bundle establishment at the time of Emx2 activation determines whether the centrioles will repolarize. Furthermore, since most HCs at E14.5 are not sensitive to microtubule disruption as in E13.5, the time-window when a nascent hair cell can respond to effects of Emx2 may be quite limited.

8) Figure 6C. Ninein staining around the centrioles at different phases of centriole migration should be quantified to demonstrate the "broad" vs. "focused" distribution pattern.

Per the reviewer’s request, we have quantified the distribution of the ninein staining during centriole migration and those results are now shown in Figure 6. On the other hand, association of ninein with a specific centriole was not consistent and those results are also included in Figure 6.

9) Figure 7 shows one example of destabilized centriole position upon Nocodazole treatment and centriole re-positioning following washout. Since treated MHCs were at different stages of polarization, it should be both feasible and informative to assess a number of treated cells and determine whether and how many (a) Phase I cells proceeded to Phase II, (b) Phase II cells reverted to Phase I (as shown in Figure 7C), and (c) post-Phase II cells with unaffected centriole position in the presence of Nocodazole. This would more convincingly demonstrate the role of microtubules in kinocilium migration.

In the revised manuscript we have now included results showing nocodazole effects on hair cells in Phase I and beginning of Phase II, in addition to the existing results of hair cells in late Phase II (see Figure 7). To address the effects of nocodazole on post-Phase II hair cells, we analyzed nocodazole-treated utricular explants at E14.5 (see Figure 8). Under the same dosage and culture conditions, we did not observe similar changes in centriole movements in hair cells at E14.5 compared to E13.5, suggesting that microtubules are more important during the centriole migration phase of hair bundle establishment.

10) Given that the role of kinocilium in utricular hair bundle polarity is debated, it would be important to interfere with microtubules and assess hair bundle polarity. Does prolonged treatment of utricle explants with either nocodazole or SiR-tubulin (which can stabilize microtubules) affect hair bundle polarity?

Prolong treatment of E13.5 utricular explants with 5 μM nocodazole for 6 hours caused the two centrioles to remain in the center of the hair cell (Figure 8A-C) similar to treatment after 1 hr (Figure 7). We extrapolated from these results that hair bundle polarity will most likely be affected due to mispositioned centrioles, even though our culture system does not last long enough for us to specifically address hair bundle polarity. However, nocodazole treatments of E14.5 cultures did not cause the two centrioles to move to the center of the hair cell as readily as in E13.5 cultures. Even though some centrioles did move towards the center eventually, the movements were quite different at E14.5, where the two centrioles moved in unison, suggesting that perhaps these movements may be secondary due to reduction of the apical surface of the hair cells (Figure 8). Therefore, these results suggest that centriole positioning in more mature hair cells is less sensitive to microtubule disruption.

11) Related to above, studies of auditory hair cells by Ezan et al. suggest that microtubule cortical capture is critical for kinocilium migration. Thus it would be important to test whether blocking microtubule dynamic instability (without destroying microtubules per se) using low-dose nocodazole would disrupt Ninein localization, Phase I and Phase II centriole behaviors and utricle hair bundle polarity.

In principle, this is a good idea. In practice, it is a very difficult experiment to execute since nocodazole basically affects microtubule polymerization and stability. It may not be feasible to find an optimal low dosage that does not affect microtubule dynamics too much but changes centriole trajectories. Nevertheless, we made an attempt and tried an experiment using 1μM of nocodazole and the centriole migration under this concentration was normal. For reasons described above, we decided to abandon this line of experiments.

12) The relationship between cuticular plate formation and kinocilium migration was not addressed and warrants further analysis. Correlating anti-spectrin staining with centriole position in hair cells at early stages of polarization (e.g. Phase I, Phase II and post-Phase II equivalent) would inform how the developing cuticular plate might physically constrain centriole migration and impede repolarization competence.

We have correlated different stages of centriole migration with formation of the cuticular plate and polarity complex LGN/Ins/Gαi by using anti-spectrin and anti-Gαi antibodies, respectively. Our results indicate that anti-Gαi staining is present during Phase I but the signal is diffused. Then, starting at early Phase II, both spectrin and Gai staining are apparent and on opposite side of the hair cell. This complementary expression pattern between spectrin and Gαi continues to strengthen and maintained by late stage of Phase II (see Figure 9). Therefore, the reviewer may be right that the developing cuticular plate may constrain centriole migration and impede repolarization competence. We have incorporated this idea into our Discussion. Thank you!

Reviewer #2:[…]The authors mention an interesting study by the group of Michel Bornens (Piel et al., 2000) where Piel and colleagues evaluate the role of the cytoskeleton on centrioles movement. These authors notably show that the "characteristic motions of the daughter centriole persist in the absence of microtubules (Mts), or actin, but are arrested when both Mts and actin filaments are disrupted." Tona and Wu use transient nocodazole treatment to disrupt MT and as a result observed a return of the centrioles to the center of the hair cell and an increase in DC movement. But is actin also participating in the peripheral movement of the centrioles and/or its maintenance at the periphery? The authors seem to suggest that it is only MT dependant.

Per the reviewer’s suggestions, we investigated the role of actin in centriole trajectory using an inhibitor of actin filament organization, latrunculin-A. We conducted two experiments using 5 μM of Lat-A for approximately 11 hours. Under these conditions, the centrioles did not readily migrate centrally as nocodazole-treated samples at E13.5. In fact, the centrioles in Lat-A-treated samples behaved more like E14.5 HCs treated with nocodazole that both centrioles appeared to move gradually towards the cell center. However, since we haven’t done a dose-response experiment, we cannot confidently conclude whether actin plays a role in centriole migration or not. Given our manuscript is already data-heavy with 10 figures and 9 supplementary figures, we feel the role of actin should be deferred for a future study.

In that same paper, the authors measured the area covered by the centriole as a measure of the persistent motion exhibited by a centriole (Figure 5C of that paper). Would it be possible, with the data already at hand, to extract an area covered by the centriole, with respect to HC surface? Or at the very least to highlight a bit better the typical trajectory of a MC and a DC? The trajectories as illustrated in Figure 1D in yellow and orange are difficult to follow. Though the corresponding videos are quite clear, it would be nice to have a schematic with an expended HC surface and different colors to better follow.

There were so many frames taken of the centriole positions that it is difficult to demonstrate the temporal trajectory well. Therefore, we decided to highlight the most critical period, in which the daughter centriole precedes ahead of the mother centriole in the migration to the peripheral location (see Figures 2, 3, 5 and 7; Figure 2—figure supplement 3 and Figure 7—figure supplement 1). We hope this addition will help to clarify the centriole movements recorded. Additionally, we included the apical surface of the hair cell by using the Imaris software, in pertinent figures as well.

I am not entirely convinced by the conclusions from Figure 6. It is a really informative piece of data but despite a number of cells recorded displaying what seems to be a similar profile of expression of ninein, the staining is spread out and uneven. I would point to Figure 6C Phase II (early), the ninein staining, though covering both centrioles, seems stronger/enriched at the level of the MC. Is it always true? Same observation for image Phase II (late). Is there a way to illustrate this with a heat map or some sort of quantification?

We have quantified the distribution of ninein in relationship to centrioles during Phase I and II. While the migration of centrioles was always consistent with a broader ninein expression domain, the location of ninein in relationship to the two centrioles was quite variable. These results are now shown in Figure 6.

The authors speculate on temporal sensitivity of the hair cells in response to Emx2 in different contexts: the transgenic knockout mice, the overexpression mice, and the AAV. They discuss this in the subsection “The role of Emx2 in reversing hair bundle orientation”, always in terms of timing. I am wondering if the levels of Emx2 could not participate in the overall mechanism. It is well known for example that core PCP signaling needs to be finely controlled, as too much or too little of core PCP protein will affect the polarity of the tissue. Could the authors speculate, based on what is known about Emx2 in other species and/or tissues, regarding this aspect? I agree that the 3 days delay between the detection of Emx2 in the future sensory epithelium and the detection of hair cells could reflect other roles of Emx2, but could it also be necessary for some accumulation? From a quick survey of the literature, it seems a concentration-dependent mechanism has been suggested for Emx2 in cortical progenitors.

We agree that an optimal concentration is probably important for most proteins in cells. However, considering the short lag time (approx. 12 hours) of AAV-Emx2 in changing the trajectory of the centrioles and yet this effect is not necessarily correlated with the high levels of ectopic Emx2 expression suggests that the onset of Emx2 expression 3 days ahead of hair bundle establishment is likely to have a different function rather than time required to accumulate enough Emx2 to mediate hair bundle orientation.

Regarding the discussion on "pulling" versus "shortening" of the MT (subsection “Hair bundle establishment in HCs”), I do not think the two are mutually exclusive and that one should prevail. Even if the authors have supporting data for the former, the latter is probably also at play. The simplest reason being that the accumulation of the MT fibers has to somehow be "evacuated" as it is pulled, to prevent its accumulation on one side of the HC.

We agree with the reviewer and the description has been modified in the revised manuscript. See the Discussion and other places of the manuscript.

Reviewer #3:[…]Immunostaining of the centrioles should be performed. In particular looking a centrin (mother and daughter), Odf2 (mother-specific), PARP3 and centrobin (daughter-enriched) should be done. Ideally this would be done with Gai staining as well, to ask if the position of the daughter centriole is coordinated.

We have tried some of these antibodies, but the staining is not optimal. Nevertheless, we are confident in our identification of the mother and daughter centrioles, based on the association with or the lack of Arl13b staining and the relative difference in the apical-basal differences between the location of the two centrioles. Furthermore, the additional Gαi staining correlated well with the position of the daughter centriole (see Figure 9) suggesting that the LGN/Ins/Gαi complex may be involved in regulating the daughter centriole migration as well. In fact, the DC being more laterally positioned than the MC was found to be closer to the LGN/Ins/Gαi complex.

Supporting cells also have centrioles that will generate the basal body for their cilia. What are the supporting cell centrioles doing? Are they coordinated as well? Does Emx2 change their migration?

The centrioles in the supporting cells, in either medial or lateral utricle, remained in the center of the cell without migration to the periphery, unlike centrioles in hair cells. (see Figure 2—figure supplement 4).

One of the obvious parallels is that with spindle orientation. It is surprising that the Gai/LGN system was not directly investigated, although the insinuation is that this data is available (in Figure 8?). What is migration like when the explant is treated with pertussis toxin.

We treated the utricular explants with pertussis toxin and we are happy to report that the in vitro effects of pertussis toxin in changing the hair bundle orientation were more profound than our previously published results using *Gfi1cre; Rosa-Ptx* transgenic utricles. These results are now included in Figure 9—figure supplement 1A. However, we did not live image the centriole migration pattern assuming that the centrioles will be following the default trajectory moving to the lateral periphery similar to the Md HCs and we don’t think these experiments will be particularly informative given all the conditions that we have already live-imaged in the study.